

# Towards an operational Ice Cloud Imager (ICI) retrieval product

Patrick Eriksson[1], Bengt Rydberg[2], Vinia Mattioli[3], Anke Thoss[4], Christophe Accadia[3], Ulf Klein[5], and Stefan A. Buehler[6]

[1]Department of Space, Earth and Environment, Chalmers University of Technology, Gothenburg, Sweden
[2]Möller Data Workflow Systems AB, Gothenburg, Sweden
[3]EUMETSAT, Darmstadt, Germany
[4]Swedish Meteorological and Hydrological Institute (SMHI), Norrköping, Sweden
[5]European Space Agency/ESTEC, Noordwijk, Netherlands
[6]Department of Earth Sciences, Universität Hamburg, Hamburg, Germany

**Correspondence:** Patrick Eriksson (patrick.eriksson@chalmers.se)

**Abstract.** The second generation of the EUMETSAT Polar System (EPS-SG) will include the Ice Cloud Imager (ICI), the first operational sensor covering sub-millimetre wavelengths. Three copies of ICI will be launched that together will give a measurement series exceeding 20 years. Due to the novelty of ICI, preparing the data processing is especially important and challenging. This paper focuses on activities related to the operational product planned, but also presents basic technical characteristics of the instrument. A retrieval algorithm based on Bayesian Monte Carlo integration has been developed. The main retrieval quantities are ice water path (IWP), mean mass height ($Z_m$) and mean mass diameter ($D_m$). A novel part of the algorithm is to fully present the inversion as a description of the posterior probability distribution. This is to prefer for ICI as its retrieval errors not always follow Gaussian statistics. A state-of-the-art retrieval database is used to test the algorithm and to give an updated estimate of the retrieval performance. The degrees of freedom in measured radiances, and consequently the retrieval precision, vary with cloud situation. According to present simulations, IWP, $Z_m$ and $D_m$ can be determined with 90% confidence at best inside 50%, 700 m and 50 μm, respectively. The retrieval requires that the data from the thirteen channels of ICI are remapped to a common footprint. First estimates of the errors introduced by this remapping are also presented.

## 1 Introduction

Satellite data are today an indispensable part of numerical weather prediction (NWP), see e.g. Bauer et al. (2015). The first observations from space directed towards weather prediction were made during the early 1960s by the TIROS (Television InfraRed Observation Satellite) program, using optical and infrared sensors (Bandeen et al., 1961). According to Staelin et al. (1976), the first satellite-based microwave observations of Earth's atmosphere were made by Cosmos 243 and 384, launched by the Soviet Union in 1968 and 1970, respectively. Atmospheric humidity and liquid cloud water were measured using channels at 22.235 and 37 GHz. These first, brief measurements (two weeks and two days, respectively) were followed by NEMS



(Nimbus E Microwave Spectrometer) that was functional 2.4 years after its launch 1972 with Nimbus-5. The channels of NEMS were placed at 22.235, 31.4, 53.65, 54.9 and 58.86 GHz. The additional channels around 55 GHz gave information on the atmospheric temperature profile (Waters et al., 1975). Another microwave sensor onboard Nimbus-5 was ESMR (Electrically Scanning Microwave Radiometer), that had a single channel at 19.35 GHz and showed that rainfall can be detected from space
(Kidd and Barrett, 1990).

More regular microwave soundings started around 1979 with the MSU (Microwave Sounding Unit) and SSM/T (Special Sensor Microwave – Temperature) sensor series. Both these instruments had only channels between 50 and 60 GHz (see e.g. Grody, 1983; Liou et al., 1981). The SSM/I (Special Sensor Microwave – Imager), introduced in 1987 had humidity, cloud liquid water and precipitation as main atmospheric targets, with channels at 19.4, 22.2, 37.0 and 85.5 GHz (see e.g. Schluessel
and Emery, 1990), and thus extended the coverage of the microwave region upwards. The next main step was taken during the 1990s with the SSM/T-2 (Special Sensor Microwave – Humidity) and AMSU-B (Advanced Microwave Sounding Unit – B) instruments that both included three channels around 183.3 GHz (see e.g. Spencer et al., 1989; Saunders et al., 1995). A main motivation for extending the coverage up to 183 GHz was to obtain vertical information on humidity, and not only column values. Today a relatively high number of microwave sensors are operational, mainly in sun-synchronous orbits but also in
other orbits, such as SAPHIR (Sondeur Atmospherique du Profil d'Humidite Intertropicale par Radiometrie) and GMI (Global precipitation mission Microwave Imager).

In NWP the capability of providing information on temperature and humidity with no or small impact of clouds has traditionally been seen as the main justification for launching microwave receivers. It has been recognised that passive microwave data also contain valuable information on clouds and precipitation and these features have been used in various stand-alone
retrievals (e.g. Spencer et al., 1989; Weng et al., 2003; Andersson et al., 2010), but this fraction of the data has been rejected inside NWP as assimilation systems have been incapable of dealing with these observations. This situation has started to change, and there are already indications of a strong increase of the relative impact of microwave data inside NWP (Geer et al., 2017).

The present growing impact of microwave data is mainly due to improved assimilation software in combination with increased computing power, but also new versions of the instruments having a higher number of channels has been beneficial.
But one limitation has remained for two decades, that operational microwave observations are so far limited to frequencies below 195 GHz. This situation will change in 2023 with the launch of ICI (Ice Cloud Imager), that will extend the coverage up to 670 GHz. ICI is one of the instruments planned for the next generation of Metop satellites, see further Sec. 2. The frequencies 195 and 670 GHz corresponds to wavelengths of 1.5 and 0.45 mm, respectively, and ICI will thus open up the "sub-millimetre region" for NWP.
The main objective of ICI is to provide data on humidity and ice hydrometeors, particularly the bulk ice mass. The advantage of using sub-millimetre observations for deriving such information was first pointed out by Frank Evans and coworkers in a series of articles (Evans and Stephens, 1995a, b; Evans et al., 1998, 1999, 2002). The initial idea was to have a sub-millimetre instrument onboard CloudSat to complement its cloud radar, but this part was later descoped.

The idea of a sub-millimetre cloud ice sounder was picked up again in a mission called CIWSIR, that was proposed to
the European Space Agency ESA as an "Earth Explorer" in 2002 and again in 2005 (Buehler et al., 2007). CIWSIR was not



selected, but ESA funded preparatory studies, that lead to a consolidated mission proposal called CloudIce for Earth Explorer 8 in 2010 (Buehler et al., 2012). It featured channels near 183.31, 243.20, 325.15, 448.00, and 664.00 GHz. Shortly thereafter, a similar sensor was also proposed for the international space station (ISS-ICE), with a reduced set of channels.

While CloudIce was not selected for Earth Explorer 8, it was taken as blueprint for the ICI instrument part of EUMETSAT
Polar System - Second Generation (EPS-SG). Its channel configuration, given explicitly in Table 1, is identical to CloudIce, except that the number of channels near 183 GHz was reduced from 6 to 3.

ICI will be the first operational sub-millimetre mission, but measurements of our atmosphere at such wavelengths already exist by other instruments, mainly by limb sounding instruments. The main objective of these instruments is to monitor gases in the strato- and mesosphere, but at their lowest tangent altitudes they perform observations that have similarities with ICI.
Retrievals of ice cloud mass have also been developed for all three sub-millimetre limb sounders launched so far; Aura MLS (Wu et al., 2006), Odin/SMR (Eriksson et al., 2007) and SMILES (Millán et al., 2013; Eriksson et al., 2014). Observations at 887 GHz were recently demonstrated by a "cubesat" mission (IceCube, Wu (2017)). The observation approach behind ICI has also been used by some airborne instruments. The pioneering instruments were MIR (Millimeter-wave Imaging Radiometer) and CoSSIR (Compact Scanning Submillimeter Imaging Radiometer) (Wang et al., 2001; Evans et al., 2005). More lately,
ISMAR (International SubMillimetre Airborne Radiometer) has been developed largely to support the preparations for ICI (Fox et al., 2017). These instruments and the associated data analysis, besides their intrinsic scientific value, provided justification for ICI in the selection process and provide useful input when designing processing algorithms for ICI.

It is expected that ICI data will be used in two main ways. In NWP the data will mainly be ingested as basic radiances; for a review of challenges, expected benefits and approaches of "all-sky" assimilation, see Geer et al. (2018). The data of ICI can also
be "inverted" in stand-alone algorithms to produce a number of geophysical quantities, see Buehler et al. (2012). The produced retrieval datasets can be of concern for short-term weather forecasting, but will likely mainly be used for different climate applications, such as the verification of global models made by the similar ice cloud products derived from limb sounders (e.g. Li et al., 2005; Eriksson et al., 2010; Jiang et al., 2012). This article describes activities performed under the auspices of EUMETSAT in preparation of a "day-one" retrieval product (i.e. the product released directly after commissioning), as well as
to provide general support for using ICI data.

The ICI instrument and its main characteristics are introduced in Sec. 2, while the following Sec 3 outlines the retrieval algorithm in focus. The expected performance is investigated in Sec. 4 using simulated data. The two final sections provide an outlook and conclusions.

## 2 The Ice Cloud Imager

### 2.1 Overview of EPS-SG

The ICI mission is part of the EUMETSAT Polar System second generation system (EPS-SG). The space segment will consist of a two-satellite architecture, referred to as Metop-SG satellite A and B. There will be three satellite pairs, where each satellite will have a nominal lifetime of 7.5 years to span a total operational lifetime over 21 years. These satellites will fly, like present



Metop, in a sun-synchronous mid-morning orbit at 09:30 local time of descending node. The altitude profile over the Earth geoid varies between 848 and 823 km (832 km mean altitude). The orbit repeat cycle will be 29 days (412 orbits per repeat cycle). The main ground-station will be Svalbard, but also McMurdo will be used to improve the timeliness of data. The ground segment includes also regional ground stations for receiving Direct Data Broadcast. See further www.eumetsat.int/
website/home/Satellites/FutureSatellites/EUMETSATPolarSystemSecondGeneration/index.html.

   ICI will be onboard of the B satellites, also carrying MWI (Micro-Wave Imager), SCA (Scatterometer), RO (Radio Occultation sounder) and A-CDS (Advanced Data Collection System). In particular, MWI is a conically scanning radiometer which observes 18 frequencies ranging from 18 to 183 GHz. All channels up to 89 GHz will observe in dual polarisation, while only vertical polarisation will be provided for higher frequencies. MWI has the same requirements for incidence angle and fore-
view observation as ICI. Combined, the MWI and ICI radiometers will provide an unprecedented set of microwave passive measurements, from 18.7 GHz up to 664 GHz.

## 2.2   The receiver package

The ICI radiometer (Bergadá et al., 2016) consists of seven double sideband front-ends, operating with local oscillator (LO) frequencies of 183.31, 243,20, 325.15, 448.00 and 664.00 GHz. The frequencies 183.31, 325.15 and 448.00 correspond to
three water vapour transitions, while 243,20 and 664.00 GHz are "window" channels (see Buehler et al., 2007, Fig. 10). There is a receiver at each of these LO frequencies providing data matching vertical (V) polarisation inside the atmosphere. At both the two window frequencies there is also a second receiver covering horizontal (H) polarisation. A spectrometer of filter-bank type is attached to each front-end. The receiver package will be kept in thermal balance by passive cooling. Presently, the receiver noise temperature is expected to be about 600, 900, 1700, 1500 and 2600 K at the five LO frequencies, respectively.
For the window frequency receivers the filter-bank consists of a single channel, while the other filter-banks have three channels each. Position and width of all the channels are reported in Table 1 and are visualised in Fig. 1.

## 2.3   Antenna system, scanning and calibration

The receiver package is integrated with a conically scanning antenna system. The diameter of the main reflector is 0.26 m (slightly elliptical), and the system is rotating at 1.333 Hz (i.e. 45 r.p.m.). Atmospheric observations are made over about 120°,
around the platform's (forward) flight direction. This gives a swath width of roughly 1500 km. The platform will perform yaw manoeuvres to keep the swath centred around the sub-nadir orbit track. During the remaining part of each rotation, calibration data will be obtained by observing "cold sky" and an internal calibration target that will have a temperature of around 300 K. The overall requirement on random (NEΔT) and systematic (bias) uncertainties of calibrated antenna temperatures are found in Table 1.
The horn antennas are designed to keep the angular resolution the same between channels (about 0.5°), but the footprints of the receivers still differ, as the antenna of each front-end is placed at a different position in the focal plane. The angular offsets are found in Table 1. The reference angle for the elevation offsets is 44.767°, measured from the nadir direction. This gives a configuration of instantaneous footprints at surface level as depicted in Fig 2, with surface incidence angles varying



**Table 1.** Specifications of the ICI receiver. ICI has double sideband receivers, indicated by $\pm$ in the third column, and the bandwidth refers to the width of single passbands, i.e. the intermediate frequency bandwidth. "NE$\Delta$T" and "Max bias" are reported as the requirements, and final performance should be better. Further comments are found in the text (the two last columns are discussed in Sec.3).

| Channel Name | ID | Frequencies [GHz] | Bandwidth [GHz] | Polari-sation | NE$\Delta$T [K] | Max bias [K] | Elevation offset [°] | Azimuth offset [°] | $\tau = 1$ [km] | Ozone [K] |
|---|---|---|---|---|---|---|---|---|---|---|
| ICI-1V | 1 | 183.31±7.00 | 2.00 | V | 0.8 | 1.0 | -0.780 | 0.000 | 0.8-3.8 | 0.1 |
| ICI-2V | 2 | 183.31±3.40 | 1.50 | V | 0.8 | 1.0 | -0.780 | 0.000 | 2.8-5.6 | 0.1 |
| ICI-3V | 3 | 183.31±2.00 | 1.50 | V | 0.8 | 1.0 | -0.780 | 0.000 | 3.8-6.8 | 0.1 |
| ICI-4V | 4 | 243.20±2.50 | 3.00 | V | 0.7 | 1.5 | 0.711 | -3.398 | 0.0-2.5 | 0.1 |
| ICI-4H | 5 | 243.20±2.50 | 3.00 | H | 0.7 | 1.5 | 0.731 | 3.385 | 0.0-2.5 | 0.1 |
| ICI-5V | 6 | 325.15±9.50 | 3.00 | V | 1.2 | 1.5 | -0.822 | -2.226 | 1.6-4.4 | 0.2 |
| ICI-6V | 7 | 325.15±3.50 | 2.40 | V | 1.3 | 1.5 | -0.822 | -2.226 | 3.1-5.9 | 0.2 |
| ICI-7V | 8 | 325.15±1.50 | 1.60 | V | 1.5 | 1.5 | -0.822 | -2.226 | 4.4-7.4 | 0.9 |
| ICI-8V | 9 | 448.00±7.20 | 3.00 | V | 1.4 | 1.5 | -0.822 | 2.240 | 4.5-7.2 | 0.1 |
| ICI-9V | 10 | 448.00±3.00 | 2.00 | V | 1.6 | 1.5 | -0.822 | 2.240 | 6.0-8.9 | 0.1 |
| ICI-10V | 11 | 448.00±1.40 | 1.20 | V | 2.0 | 1.5 | -0.822 | 2.240 | 7.2-10.2 | 0.3 |
| ICI-11V | 12 | 664.00±4.20 | 5.00 | V | 1.6 | 1.5 | 0.752 | -1.367 | 4.5-7.1 | 1.6 |
| ICI-11H | 13 | 664.00±4.20 | 5.00 | H | 1.6 | 1.5 | 0.875 | 0.941 | 4.5-7.1 | 1.6 |

between 51.5° and 53.8°. The instantaneous footprint sizes at surface level are about 17/20 km along-track and 7.3/8.5 km across-track for the footprints having a positive/negative elevation offset (at -3 dB, and slightly varying with latitude). The angular movement inside the integration time increases the effective across-track size.

Although the combination of conical scanning and the platform's movement in total gives a continuous coverage over the

5  swath, there will not be any perfect matches in horizontal coverage between the channels. Accordingly, some post-processing is required to obtain data suitable for an inversion using channels from more than one front-end. To support footprint "remapping" a high across-track sampling will be applied, data will be recorded every 0.661 ms. This corresponds to an across-track movement of the boresights between samples of about 2.7 km, giving 784 samples/scan. The distance along-track between subsequent scans will be about 9 km. This gives substantial overlap of sample footprints, both in along- and across-track di-

10  mension, giving some freedom in setting the target resolution in the remapping of footprints. The requirement on final footprint size is 16 km (as average between along- and across-track resolution), and the requirement of e.g. NE$\Delta$T is defined for this horizontal resolution. The noise in individual samples will be higher. It is expected that averaging over four subsequent across-track samples will meet the requirements, and about 200 footprints/scan will effectively be provided. L1b data will only contain the original samples, the optimal remapping will differ depending on application.



# 3 Algorithm

## 3.1 Aim and constraints

The planned output of the EPS-SG Overall Ground Segment at EUMETSAT Headquarters includes the MWI-ICI-L2 product, that will contain retrievals based on MWI and ICI and be delivered in near real time. The objective of the IWP product of MWI-ICI-L2 is to provide a day-one, robust, retrieval that reflects the main information content of ICI radiances. For some centrally generated level 2 products, the EUMETSAT Satellite Application Facilities (SAFs) provide support by specifying the level 2 processing algorithms and share responsibility for the products. The SAF supporting Nowcasting (NWC-SAF) retains the scientific ownership of the IWP product and delivered the IWP algorithm theoretical basis definition (Rydberg, 2018). To allow for the procurement and implementation in the ground segment, the IWP algorithm definition had to be finished during 2018, with further changes in the algorithm specifications not to impact the basic architecture and design. Additional products from ICI will be generated directly by the SAFs located at weather services in EUMETSAT member and co-operating states.

## 3.2 Overview

A first, crucial decision was the selection of retrieval approach. "Optimal estimation" (a.k.a. 1DVAR) was not selected as it would demand a forward model handling multiple scattering of polarised radiation and capable of providing the Jacobian with respect to the retrieval quantities. Such a model was simply not at hand. With respect to sub-millimetre cloud observations, optimal estimation has so far only been used for theoretically inclined studies (Birman et al., 2017; Grützun et al., 2018; Aires et al., 2019).

Further, the retrieval problem at hand is both non-linear and involves non-Gaussian statistics, and a more general solution of the Bayes theorem should be preferable. For practical reasons this leads to approaches based on a retrieval database (Rydberg et al., 2009). The most straightforward implementation can be denoted as BMCI (Bayesian Monte Carlo Integration), and has been the method of choice for Evans and coworkers (e.g. Evans et al., 2002).

There are close connections between BMCI and the standard use of neural nets (Pfreundschuh et al., 2018). Such neural nets (NN), a form of machine learning, have been applied on both simulated ICI data (Jimenez et al., 2007; Wang et al., 2017) and ISMAR field data (Brath et al., 2018). Both approaches (BMCI and NN) were considered initially, but NN was eventually rejected as it was found that a very high number of nets would be required and there was no established way to estimate retrieval uncertainties.

Following the selection of BMCI, a complete retrieval algorithm was designed (Fig. 3). The algorithm consists of two main parts, a series of pre-processing steps and the actual inversion by BMCI. Only the most critical aspects are discussed in the following sections, for details we refer to Rydberg (2018). The generation of the final retrieval database is a task of the future, but a possible manner to generate the database is still outlined in Sec 4.2.



### 3.3 Input and output

The main input to the retrieval algorithm are geo-located and calibrated antenna temperatures, i.e. L1b data. Data from a number of footprints will be involved in each inversion, being remapped to the target footprint specified (Sec. 3.4.1). The target footprint also governs the extraction of geophysical variables (Sec. 3.4.3). All important retrieval parameters are set by

a configuration data structure.

The main output variables (L2) are ice water path (IWP), mean mass height ($Z_m$) and mean mass diameter ($D_m$). All these three variables are reported as percentiles of the estimated a posterior distribution (Sec. 3.5.1) and are defined as antenna weighted means. For example, the reported IWP is an estimation of

$$\text{IWP} = \frac{\int_{z_0}^{\infty} \int_{\Omega} r(\Omega) \text{IWC}(x(\Omega), y(\Omega), z) \, d\Omega \, dz}{\int_{\Omega} r(\Omega) \, d\Omega}, \tag{1}$$

where $r$ is the antenna pattern, $\Omega$ is solid angle, $x$, $y$ and $z$ are Cartesian coordinates and IWC is ice water content:

$$\text{IWC} = \int_0^{\infty} n(d_{\text{veq}}) m(d_{\text{veq}}) \, dd_{\text{veq}}, \tag{2}$$

where $n$ is particle size distribution, $m$ is particle mass and $d_{\text{veq}}$ is equivalent volume diameter ($\rho$ is the density of ice):

$$d_{\text{veq}} = \sqrt[3]{6m/\pi\rho}. \tag{3}$$

The start of the altitude integration in Eq. 1, $z_0$, is presently set to be the surface altitude, but it can be changed.

Mean mass height is defined as

$$Z_m = \frac{\int_{z_0}^{\infty} z \int_{\Omega} r(\Omega) \text{IWC}(x(\Omega), y(\Omega), z) \, d\Omega \, dz}{\text{IWP}}, \tag{4}$$

and mean mass size as (cf. e.g. Delanoë et al., 2014, Eq. 3)

$$D_m = \frac{\int_0^{\infty} d_{\text{veq}}^4 \int_{z_0}^{\infty} \int_{\Omega} r(\Omega) n(d_{\text{veq}}) \, d\Omega \, dz \, dd_{\text{veq}}}{\int_0^{\infty} d_{\text{veq}}^3 \int_{z_0}^{\infty} \int_{\Omega} r(\Omega) n(d_{\text{veq}}) \, d\Omega \, dz \, dd_{\text{veq}}}. \tag{5}$$

The L2 data will contain further data, such as retrieved water vapour column, but the exact L2 format is not finalised and only

the three main retrieval quantities are discussed below.

### 3.4 Pre-processing part

#### 3.4.1 Target footprint and remapping of data

The exact geo-location of samples differs between channels (Sec 2.3), but the time integration of individual samples is shorter than the time period necessary to sweep out a single projected field of view. This allows for a footprint matching procedure by

remapping of the original data. A toolbox for performing such remappings has been developed in a dedicated study issued by EUMETSAT (Rydberg and Eriksson, 2019). The toolbox is based on the Backus-Gilbert methodology (Backus and Gilbert,



1970; Stogryn, 1978), that earlier has successfully been applied for footprint-matching between various satellite data (e.g. Bennartz, 2000; Maeda and Imaoka, 2016).

In short, the Backus-Gilbert methodology can be used to obtain a set of optimal weighting coefficients for neighbouring samples, both within the scan and from adjacent scans, to create a remapped representation of the data matching a specified target footprint. A remapped value is a linear weighted combination of data of the channel of concern. The weights are found, after a trade-off analysis, by minimisation of a penalty function that considers both the effective noise of the remapped data and the fit to the target footprint.

The centre position of a retrieval is set by selecting one of the sample footprints of ICI-1V. The exact shape of the target footprint around this position will be determined later, but it is expected to be $\approx 16\,\mathrm{km}$ semi-circular. The effective noise of remapped samples should be equal or below the "NE$\Delta$T" reported in Table 1. Example results are found below, in Sec. 4.1.

### 3.4.2 Bias correction

The algorithm allows for a simple "bias correction" of the data:

$$T_{a,j}^c = a_j + b_j T_{a,j} \tag{6}$$

where $T_{a,j}^c$ is corrected antenna temperature for channel $j$, $T_{a,j}$ is the value as given by the remapping toolbox and $a_j$ and $b_j$ are channel specific coefficients.

The purpose of the bias correction is to remove systematic differences between remapped L1b data and the simulations behind the retrieval database. A bias can originate from e.g. calibration issues, the remapping and incorrect spectroscopic data in the simulations. This module will only be applied as a rough temporary solution if any bias is detected, until the source to the bias has been understood and corrected.

### 3.4.3 Geophysical data and RTTOV

The retrieval performance can be improved by incorporating various geophysical data. These data will be taken from the ECMWF forecast system. Data of dynamic character that will be used include: temperature, ozone and surface wind speed, while static data are various parameters to characterise surface altitude and type. The water vapour profile is also imported from ECMWF, but it is modified below the tropopause to have a constant relative humidity (a configuration setting). The logic behind this approach is to incorporate information on e.g. atmospheric temperatures and ozone from ECMWF (ICI has no temperature channels), while letting humidity be constrained by the ICI data. The last column in Table 1 gives the mean impact of ozone based on a set of simulations. The maximum impact found was 2.1 K, for ICI-11 and a mid-latitude winter scenario.

Using the ECMWF data as input, radiative transfer calculations will be performed applying the RTTOV software (Saunders et al., 2018), to obtain a first estimate of the atmospheric optical thickness and a reference antenna temperature ($T_a^r$). These calculations assume "clear-sky" conditions (i.e. no impact of hydrometeors), are run for all ICI channels and are discussed further below.



### 3.4.4 Channel selection

Modelling of surface effects will, at least initially, be a main obstacle for these retrievals. Simulating these effects for land surfaces is a challenge already at low microwave frequencies. The situation for water bodies is better, particularly as the TESSEM sea-surface emissivity parameterisation has been updated to cover the full frequency range of ICI (Prigent et al., 2017). Some validation of TESSEM has been made (using ISMAR), but presently relatively large model uncertainties are expected even for water surfaces.

The impact of surface effects on measured radiances depends mainly on the atmospheric transmission. The transmission varies strongly between the ICI channels, as exemplified in Fig. 4. It varies also with the atmospheric situation. Estimates of at what altitude the transmission to ICI equals $e^{-1}$, for clear-sky conditions, are found in the column "$\tau = 1$" of Table 1. The lowest altitudes are associated with driest atmospheric scenario considered, and vice versa. The table shows that surface effects is in general of no concern for e.g. ICI 7V, 10V, 11V and 11H, while for some channels the surface must always be considered.

As a consequence, an adaptive selection of data is required. A channel mask is formed by evaluating

$$\tau_{\mathrm{cs},j} + c_{\mathrm{hm}}\tau_{\mathrm{hm},j} \geq \tau_t^s \tag{7}$$

where $\tau_{\mathrm{cs},j}$ is the clear-sky optical thickness of channel $j$ obtained by RTTOV, $c_{\mathrm{hm}}$ a configuration setting, $\tau_{\mathrm{hm},j}$ is estimated additional optical thickness due to hydrometeors and $\tau_t^s$ is a threshold value for surface type $s$. Data from channels fulfilling this criterion are included in the calculations. In the pre-processing part $\tau_{\mathrm{hm}}$ is set to zero. The channel mask is re-evaluated as part of the BMCI module, then also including attenuation due to hydrometeors. Both $c_{\mathrm{hm}}$ and $\tau_t^s$ are configurable variables, where the later is specified for five different surface types.

### 3.4.5 Detection of clear-sky data

The algorithm includes an optional module for identifying observations that with a high probability match clear-sky conditions, that thus can be set to give $\mathrm{IWP} = 0$ without doing an actual inversion. This procedure results in that the L2 structure can not be fully filled, e.g. the water vapour column will not be retrieved, and this module will only be activated if it will be necessary to decrease the overall calculation burden of the processing. As the module likely will not be applied, no details are here given.

### 3.5 Inversion part

### 3.5.1 Theory and retrieval representation

The retrieval is performed by the BMCI method (Sec. 3.2). For a description of BMCI and its relationship to Bayesian estimation, see e.g. Kummerow et al. (1996) or Pfreundschuh et al. (2018). In short, BMCI is based on a "retrieval database" consisting of $n$ pairs of atmospheric state, $x_i$, and corresponding observation, $\boldsymbol{y}_i$, with the constraint that $x_i$ is approximately distributed according to reality, i.e. represents the prior distribution of $x$. The essence of BMCI is, for a given measurement $\boldsymbol{y}$,





to attribute a posterior probability, $p(x_i|\boldsymbol{y})$, to each database state as

$$p(x_i|\boldsymbol{y}) = w_i a_i / \sum_{i=1}^{n} w_i a_i, \tag{8}$$

where $w_i$ is a measure on the agreement between $\boldsymbol{y}$ and $\boldsymbol{y}_i$,

$$w_i = \exp(-\left[(\boldsymbol{y}-\boldsymbol{y}_i)^T \boldsymbol{S}_o^{-1}(\boldsymbol{y}-\boldsymbol{y}_i)\right]/2), \tag{9}$$

with $\boldsymbol{S}_o$ being the covariance matrix describing observation uncertainties. The factors $a_i$ can be seen as a priori weights. They are not standard, but are introduced to allow tailoring of the retrieval database to the specifics of the retrievals of concern. For example, it could be justified to accept cases with IWP = 0 only with some probability $r < 1$ during database generation (Sec. 5). If this thinning is performed, remaining database cases having IWP = 0 will obtain $a_i = 1/r$ (instead of 1).

The actual solution of BMCI is the estimated posterior distribution (as for all Bayesian methods), but it is unpractical to
report sets of $p$. Some more compact description is needed. If the posterior distribution follows a Gaussian distribution it suffices to report the expectation value and the width of the distribution. ICI retrievals do not fall into this category and it was decided to instead use a more general description based on the cumulative distribution function, in the continuous case defined as

$$F_{x|\boldsymbol{y}}(x) = \int_{-\infty}^{x} p(x'|\boldsymbol{y})\mathrm{d}x', \tag{10}$$

and in the framework of BMCI obtained by summing $p_n$ for all cases having $x_i < x$:

$$F_{x_i|\boldsymbol{y}}(x) = \sum_{x_i < x} p(x_i|\boldsymbol{y}). \tag{11}$$

Using Eq. 11, $F_{x|\boldsymbol{y}}$ is calculated on a wide grid of $x$-values. These data are then used to obtain the inverse distribution function, $F^{-1}$, numerically by interpolation to a set of fixed percentiles. A more descriptive name of $F^{-1}$ is the quantile function. For example, $F^{-1}(0.5)$ is the median and the 90th percentile is $F^{-1}(0.9)$. Figure 5 exemplifies prior and posterior quantile
functions.

It is presently planned to report the 5th, 16th, 50th, 84th and 95th percentiles in the L2 data. If the retrieval must be condensed to a single value, the first candidate to "best estimate" should be the 50th percentile. The other percentiles can be used in different ways. For example, if the 5th percentile for IWP is $> 0$ then a correct detection of ice hydrometeors is highly probable. The 16th/84th percentile range matches $\pm 1\sigma$ for a Gaussian distribution. The true value is between the 5th and 95th
percentiles with a probability of 90%, etc.

### 3.5.2 Measurement vector and uncertainties

The measurement vector ($\boldsymbol{y}$) incorporates data from channels fulfilling the optical thickness criterion of Eq. 7 as a difference:

$$\Delta T_{a,j} = T_{a,j}^c - T_{a,j}^r \tag{12}$$





where $T_{a,j}^c$ is defined by Eq. 6 and $T_{a,j}^r$ is a simulated antenna temperature (by RTTOV, Sec. 3.4.3). To match this, the retrieval database contains both a full (all-sky) simulation and one (clear-sky) matching $T_{a,j}^r$.

The matrix $\boldsymbol{S}_o$ (Eq. 9) shall represent both instrument and simulation uncertainties. It is kept diagonal in lack of relevant information on uncertainty correlations between channels, but also for calculation efficiency reasons. The variances $\sigma^2$ are set

as

$$\sigma_j^2 = \mathrm{NE\Delta T}_j^2 + (\Delta\epsilon T_{\mathrm{skin}}e^{-\tau_{e},j})^2 + (c\Delta T_{a,j})^2, \tag{13}$$

where $\mathrm{NE\Delta T}$ is uncertainty due to thermal noise and calibration. The second term aims at representing impact of unknown surface emissivity, where $\Delta\epsilon$ is emissivity uncertainty, $T_{\mathrm{skin}}$ is the ECMWF surface skin temperature and it is assumed that the emissivity is relatively high (impact through reflection of down-welling radiation neglected). The last term covers uncertainty

in modelling of hydrometeor scattering, where it is assumed that the modelling error is proportional to the deviation from the clear-sky reference simulation. $\mathrm{NE\Delta T}$ for each channel ($j$), $\Delta\epsilon$ for water and land, and $c$ are constants, part of the configuration data.

### 3.5.3   Database extraction and iterations

Not all database cases are included in the BMCI summation, a filtering is done based on surface type, pressure, wind speed

and temperature, as well as $\Delta T_a$. Wind speed is effectively only considered over water. The database extraction is done in an iterative manner, where the filter limits are adjusted with an iteration counter, in order to fetch both the most relevant and a sufficient number of matches.

An additional iteration scheme has been added around the core BMCI calculations. A first reason is to better make use of the observations in situations with significant hydrometeor contents. The optical thickness associated with hydrometeors is

estimated alongside of the L2 data in each iteration. Based on this updated estimate of the total optical thickness, Eq. 7 is reevaluated for all channels. If this results in that more channels can be included, BMCI is reiterated with the new channel mask. This iteration is important as the channels sensitive to the surface in a clear-sky situation, and thus ignored in the initial iteration, are the most important ones to obtain good estimates at high IWP.

The second reason is to handle the fact that the retrieval database only provides a discrete coverage of the distribution of $\boldsymbol{y}$. If

one $\boldsymbol{y}_i$ happens to agree closely with $\boldsymbol{y}$, one $w_i$ can be orders of magnitude bigger than all other $w$ and the summation in Eq. 8 will be dominated by one database case. While the median value found can be realistic, this results in an underestimation of the retrieval uncertainty. It could also be the case that no $\boldsymbol{y}_i$ gives a significant match with $\boldsymbol{y}$. Both these situations are primarily handled by increasing the variances in $\boldsymbol{S}_o$, effectively making the "search radius" larger. If this does not suffice, channels will be rejected until an acceptable number of significant weights are obtained.

For further details of the filtering and iteration schemes, see Rydberg (2018). All critical parameters are part of the configuration data.



## 4   Performance tests

### 4.1   Remapping of data

Samples from all ICI channels will be convolved into the field of view of ICI-1V. This section summarises the main findings obtained by applying the Backus-Gilbert toolbox developed (Sec. 3.4.1) on detailed radiative transfer simulations. A bias-free
convolution has been demonstrated as long as the remapping does not involve a change in incidence angle. However, this is strictly true only for ICI-2V and ICI-3V. These two channels share bore-sight with ICI-1V, but the antenna patterns differ somewhat and a remapping is still required.

Figure 6 exemplifies the issues that appear for the other channels, having an incidence angle that differs from the one of ICI-1V (Table 1). Considerable remapping errors are found for ICI-11V. The elevation offset of this channel deviates to ICI-1V
with $1.53°$, that scales to a $\sim 2$ degree lower incidence angle at surface level. This angular difference results in a remapping error even in the absence of hydrometeors, as exemplified by the upper-left portion of the simulated area. The remapping generates data that are $0.4 \text{ - } 0.6\,\text{K}$ too warm, as the toolbox can not compensate for the original difference in incidence angle. The "clear-sky" brightness temperature is higher at a lower incidence angle.

The same effect can be noted for areas with relative homogeneous cloud distributions, but brightness temperatures vary
more strongly with incidence angle in cloudy conditions and the error for ICI-11V is here instead about $0.5 - 2\,\text{K}$ (see e.g. the area directly south of $0°\text{N }10°\text{E}$). Further, at the edges of areas with hydrometeors even higher errors can be noted, as well as errors of opposite sign. These errors originate in horizontal inhomogenities. The target footprint is defined at the altitude of the reference ellipsoid. This means that line-of-sights of observations and the corresponding ones after remapping cross at the ellipsoid but deviate at altitudes inside the atmosphere. The horizontal distance between the two line-of-sights increases
with altitude, being about $700\,\text{m}$ at $12\,\text{km}$ for ICI-11V. Hence, the noted errors match the change in brightness temperature for horizontal shifts of that order. However, the atmospheric data used in the simulations do not have this high horizontal resolution and the magnitude of these errors are just indicative.

The errors found for ICI-5V (Fig. 6) show a similar spatial pattern, but have the reversed sign and are of lower magnitude. This is expected as the zenith offset of ICI-5V is only $0.04°$ smaller than the one of ICI-1V, in contrast to the larger, positive
shift for ICI-11V.

### 4.2   Generation of retrieval database

Retrieval databases for ICI must so far be generated by radiative transfer simulations. The input to the simulations can be obtained from atmospheric models providing a sufficient detailed description of hydrometeors (Wang et al., 2017; Brath et al., 2018). This approach relies on that the model mimics reality with sufficient accuracy, as it represents the a priori for the BMCI
retrieval. Another option is to base the simulations directly on observations as far as possible. As the spatial resolution of ICI is limited, the most important input is information on vertical and horizontal structures in hydrometeor fields. Today such data are available through cloud radars, even on global scale by the CloudSat one (Stephens et al., 2002).





The cloud radar data can be used in various ways. Some options are explored by Evans et al. (2002, 2012), while the results reported below are based on the methodology developed in Rydberg et al. (2007, 2009). The basic idea is to produce simulated passive observations that are consistent with the basic information provided by the radar, i.e. measured reflectivities. This is done for some assumption on particle size and shape distributions. That is, external retrievals of e.g. IWC are not

involved, the mapping from radar reflectivities to particle optical properties at the frequencies of the passive data is done by an internal, implicit retrieval. The retrieval database should contain simulations for a set of different particle assumptions, to reflect the variability and our limited knowledge of particle shapes and sizes. Remaining atmospheric data can be taken from some analysis (such as ECMWF's ERA5), but this still represents a drawback of the approach as consistency is not guaranteed. Most importantly, corrections are likely required to avoid improbable relative humdities where hydrometeors are present.

### 4.3   ICI retrieval performance

#### 4.3.1   Test retrieval database

At this stage, the simulations are based on stretches of CloudSat data. That is, the simulations have two dimensions, vertical and along-track. The ICI slant geometry and antenna pattern are represented fully inside this 2-d geometry. So far the extension of the antenna pattern in the across-track dimension is neglected, but can be included by mapping the CloudSat data to three

dimensions (Rydberg et al., 2009). Consideration of the antenna pattern is required to avoid systematic modelling biases due to "beamfilling" (Davis et al., 2007). The radiative transfer calculations were performed with the ARTS software (Buehler et al., 2018), using its interface to the RT4 (Evans and Stephens, 1995b) scattering solver. The microphyscial models applied are described in Table 2. For mid and high latitudes also simulations with a modified gamma distribution (for two habits) were produced, but the resulting $D_m$ was found to be unrealisticly high and this part of the database is here rejected. Oriented and

melting particles are so far ignored.

Observations over both water and land were simulated. Ocean surface emissivity was modelled according to Prigent et al. (2017), while land emissivity was simply set to vary randomly around 0.8 in lack of a proper model. The data used below contain in total $6.2 \cdot 10^6$ cases.

Besides the retrieval database, some data were also simulated for channels 16 - 21 of ATMS (Weng et al., 2012) and a

statistical comparison to actual observations was made. Example results are displayed in Fig 7. The peak in the distribution around 255 K corresponds to "clear-sky" situations (low level cloud can still be present), while most cases below ∼230 K should contain influences of ice hydrometeors. The agreement between simulations and observations is high down to about 200 K. For lower brightness temperatures the simulations show higher occurrence rates than the observations. This deviation is at least partly a consequence of that the full antenna pattern and particle orientation are not yet considered in the simulations.

The better agreement for nadir simulations, where ATMS has a smaller footprint, indicates the impact of the first of these two effects. The approach behind the database generation reproduces GMI data in a similar manner, even when focusing on the tropical Pacific where deep convective systems control the impact of ice hydrometeors on ICI and the radiative transfer simulations are especially challenging (Ekelund et al., submitted manuscript).





**Table 2.** Combinations of particle size distribution and habit model included in the test retrieval database. McFarquhar and Heymsfield (1997) is shortened to MH97. Field et al. (2007) defined a tropical and a mid-latitude version of their size distribution, and both are used. Each habit model consists of single scattering data selected from Eriksson et al. (2018), specified as the data's id-number inside that database. For example, 15+20 stands for a mix between thick plates (id 15) and large plate aggregates (id 20).

| Hydro-meteor | Size distribution | Habit model | Latitude region |
|---|---|---|---|
| Ice | Field et al. (2007) | 3 | Tropics |
| Ice | Field et al. (2007) | 1 | Tropics |
| Ice | MH97 | 15+20 | tropics |
| Ice | Field et al. (2007) | 1 | Mid and high |
| Rain | Abel and Boutle (2012) | 25 | Global |

A similar comparison is found in Fig. 13 of Geer and Baordo (2014). They obtained a poorer agreement with observations, with an underestimation starting at about 225 K. Similar particle models were used and the better agreement found here is likely a consequence of that the simulations are based on CloudSat, and not model data. The agreement is similar for the other ATMS channels considered, see Rydberg (2018). A graphical manner for exploring if the retrieval database covers the
multi-dimensional space spanned by the observations to be inverted is found in Brath et al. (2018, Fig. 2).

### 4.3.2   Degrees of freedom

As an introduction to the information provided by ICI, Fig. 8 displays an estimate of the measurements' degrees of freedom (DOF). The DOF can be seen as the effective number of channels. For very low IWP and most wet atmospheres, the DOF is only two. For these conditions, ICI is primarily sensitive to humidity in the middle and upper troposphere. The DOF increases
with decreasing IWV, as humidity at lower altitudes then gets a growing impact. The DOF is here about three, consistent with the fact that ICI has three channels around each water vapour transition covered (1V-3V, 5V-7V and 9V-11V, respectively), and that there is a high redundancy in information between these groups of channels (which together give an improved precision for water vapour retrievals). For most dry atmospheres, there is also a contribution to the DOF from the surface, mainly by channels 4V and 4H.
The DOF is considerably higher at high IWP. The maximum DOF in Fig. 8 is eight, but the true number is likely higher. The figure is based on simulations only including totally random particle orientation and the full information given by the dual polarisation channels is not reflected. The simulations lack also melting particles and still use a relatively low number of particle models, and the full variability of hydrometeors is probably not yet reflected.
There is an intermediate range, extending between about 10 and 500 g/m$^2$, where DOF is increasing with IWP. This analysis
shows that ICI acts mainly as a coarse humidity sounder for IWP below $\sim$10 g/m$^2$, but, as designed, provides more rich



data with increasing ice hydrometeor content. This indicates that ICI is suitable for measuring IWP, but the DOF gives no information on retrieval precision or if other quantities also can be constrained.

### 4.3.3 Overall performance

The retrieval performance was estimated by repeatedly dividing the data generated between a retrieval database and test data (Fig. 9). Since particle orientation is not yet included, these retrievals did not include the extra 243 and 664 GHz channels that measure H. Noise was added following the NE$\Delta$T of Table 1, but present tests indicate that lower noise will actually be achieved. Both these aspects should lead to a conservative estimate of the performance at low IWP, or compensating for error sources not yet considered. The results in Fig. 9 are averages of retrievals over both water and land.

The best performance is found for tropical conditions where IWP above about $50\,\mathrm{g/m^2}$ can be retrieved with good accuracy. ICI provides information also for lower IWP, down to about $10\,\mathrm{g/m^2}$, but then with an increasing influence of a priori information causing a low bias. This bias occurs because the a priori is dominated by cases having IWP=0. Accordingly, the bias could be decreased strongly by an independent method of cloud detection, effectively removing all, or most, IWP=0 from the a priori distribution.

The retrieval precision in Fig. 9 is reported as the range between the 5th and 95th percentile. This range corresponds to a 50% uncertainty above about $200\,\mathrm{g/m^2}$. The precision is poorer for lower IWP, particularly on the 5th percentile side. This percentile reaches IWP=0 when the true value is $\sim 15\,\mathrm{g/m^2}$. Mean altitude, $Z_m$ is well estimated over its full range (for the type of ice clouds of concern for ICI), i.e. between about 4 and 12 km, with a precision in the order of 700 m. The retrieval of $D_m$ is best between 175 and 400 µm, where the precision is about 50 µm, but the retrievals should be competitive between about 100 and 800 µm.

Results for mid-latitude winter conditions are also found in Fig. 9. There are likely highly uncertainties in these simulations and these results should be approached with more care. Compared to tropical conditions, the performance is poorer, especially for IWP below $100\,\mathrm{g/m^2}$. This is the case because the ice hydrometeors here are found at lower altitudes, often below the sounding range of the high frequency channels. Low IWP is best estimated by the 664 GHz channels, but they have sensitivity only down to about 5 km (Fig. 4). Low altitude clouds also make the choice of $\tau_t^s$ (Eq. 7) critical. For these test retrievals, it was set to 1 for oceans and 3 for all other surface types. $Z_m$ is retrieved without any significant bias between 2 and 10 km, but the posterior distribution is highly skewed below 3 km. That is, the 50th percentile is in general a good estimate, but the retrieval can not fully rule out considerably higher $Z_m$. The accuracy is good for $D_m$ between 150 and 600 µm, while there is a quickly growing high low bias above 650 µm.

These results do not deviate significantly from earlier similar studies. The most similar one is Jimenez et al. (2007), particularly as it also used IWP, $Z_m$, and $D_m$ as retrieval quantities. They found a better retrieval performance for low IWP, that is likely a consequence of that smaller retrieval databases were used and less error sources were considered. Our results should be more realistic, albeit possibly made in a conservative manner as explained at the start of this section. Wang et al. (2017) made a study focusing on relatively severe weather over Europe and obtained similar IWP accuracy as reported here. They did not consider $Z_m$ and $D_m$, but retrieval of separate hydrometeor classes as well as joint inversion of data from MWI and ICI.





When comparing results between studies, the error range considered must be noticed. We use a more broad range (matching $\pm 2\sigma$) compared to most others.

### 4.3.4 Test inversions

As a general sanity check, the algorithm has been been tested practically on data from some ISMAR flights. Some modifications
were needed to handle the ISMAR observations. The main additional obstacles are that the flight altitude and observation angle vary, and the retrieval database must contain simulations for all combinations of altitude and angle considered. A quantitative assessment of these retrievals was not possible due to lack of any independent estimates of IWP, but we judged the results as reasonable. At least no obvious flaws were found. These various results have been presented at conferences (e.g. Eriksson et al., 2016) and the details are not repeated here.

## 5 Outlook

The basic algorithm will not be modified until some time after the launch of ICI and the main concern for the coming years is to refine the retrieval database generation. A required extension is to include particle orientation. The impact of melting particles should be assessed and be included if found relevant. However, data on single scattering properties are lacking for both these considerations. The simulations should of course make use of most recent studies on particle size distributions and particle
shapes, preferably applying data tailored for each cloud type of concern. ISMAR should be an essential tool for validating microphysical assumptions. A first study of this type has been performed (Fox et al., 2019).

A more detailed treatment of the full antenna pattern is needed. This will increase the calculation burden, but to what extent is not yet known. The present assumption is that an independent beam approximation (IBA) can be applied, i.e. that the radiance at one location can be sufficiently well estimated by a simulation of one-dimensional character. However, test simulations have
revealed that this is not true for all situations but full three-dimensional, polarised simulations can so far only be performed by computationally costly Monte Carlo methods. For this reason IBA would be to prefer. The error by applying IBA is being assessed as part of a EUMETSAT fellowship project.

As discussed in Sec. 4.1, the necessary spatial remapping of channels causes some errors due to the differences in incidence angle. These remapping errors must either be incorporated in the generation of the database or be treated as an observation
uncertainty. In the later case, an error model must be derived to set $\boldsymbol{S}_o$ (Eq. 9) accordingly. The information on temperature and ozone obtained from ECMWF (Sec. 3.4.3) has uncertainty and the resulting impact on the retrievals has not yet been studied. The same is true for errors in assumed spectroscopic parameters, used to calculate the absorption due to gases. As ICI will operate in a relatively unexplored wavelength region, considerably spectroscopic uncertainties can not be ruled out at this point (Mattioli et al., 2019). Also here ISMAR should be an useful tool for validation.
A number of retrieval configuration settings need to be determined. For example, the optical thickness thresholds ($\tau_t^s$, Eq. 7) should be reevaluated at some point, then preferably with improved knowledge, obtained by ISMAR, of the variability of

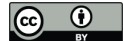



surface emissivity at the frequencies of ICI. Another example is that a clear strategy for the database thinning discussed in Sec. 3.5.1 is lacking, only rudimentary tests have so far been made.

In a longer perspective, joint inversions of data from MWI and ICI shall be considered. Such synergistic retrievals should be especially beneficial for obtaining consistent data on liquid and ice hydrometeor properties (Wang et al., 2017). The remapping
toolbox is prepared to handle this extension, but application of BMCI becomes more problematic as dealing with the combined measurements drastically increases the required retrieval database size. Machine learning could be an alternative. In Pfreundschuh et al. (2018) it is shown that quantiles of the posterior distribution can be estimated by neural networks more efficiently than with BMCI.

## 6   Conclusions

Ice hydrometeors constitute one of the components in Earth's atmosphere least constrained by present observation and modelling systems. There is even a persistent large spread among zonal means of IWP (Waliser et al., 2009; Eliasson et al., 2011; Duncan and Eriksson, 2018). ICI will provide observations that could be used to decrease these uncertainties both inside weather forecasting and stand-alone retrievals, as well as by model verification through "satellite simulators". ICI does not offer the spatial resolution of cloud radars, such as the CloudSat one, but has the swath width needed for obtaining semi-global
coverage on a daily basis.

The focus of this article is the ICI retrieval algorithm (Rydberg, 2018) that will be applied operationally at EUMETSAT. At the time of the algorithm selection, BMCI was judged a safer option than existing machine learning alternatives. However, since machine learning is developing rapidly, future scientific retrieval algorithms may well employ it.

The "day-one" algorithm described here aims at extracting the basic information of ICI on ice hydrometeors, which is the
ice water path, as well as cloud altitude and particle size. ICI has the potential to also provide profiles of ice water content (Wang et al., 2017; Birman et al., 2017; Grützun et al., 2018; Aires et al., 2019), but that possibility is so far left for research groups to investigate.

An innovative aspect of the new algorithm is that, to our best knowledge, it is the first example where the retrieval result is presented fully as a description of the posterior distribution (by reporting five percentiles), and not as the expectation value and
some uncertainty value. This more general approach is to prefer for ICI as the retrieval uncertainty can exhibit a highly skewed distribution.

The core algorithm has successfully been tested using ISMAR and simulated ICI data, but the final retrieval performance is mainly determined by the quality of the retrieval database provided to the processing system. Such a database has been produced for test purposes and to provide updated estimates of the retrieval precision. The database reflects the state of the art,
but the retrieval error estimates should still be considered as tentative because some tools needed to cover the full complexity of the observations are still lacking.

It is hard to find stringent uncertainty estimates of other IWP retrievals, but we note that the global mean of IWP given by the DARDAR inversions (mainly based on CloudSat) changed by 26% between the two most recent versions (Cazenave et al.,



2019). Based on present simulations, ICI will deliver a similar accuracy at least above IWP $= 200\,\mathrm{g/m^2}$. Above this IWP, there is no intrinsic cause of bias in the retrievals, and the precision for single retrievals is $\pm 50\%$ (at quantiles matching $\pm 2\sigma$).

The use of ICI in numerical weather prediction (NWP) is not discussed here, but several activities described are also relevant for this application. The most notable example should be the development of ice hydrometeor single scattering data (Eriksson
et al., 2018), that is of direct relevance for "all-sky" assimilation of ICI radiances. A problem common for NWP and stand-alone retrievals is how to incorporate the effect of ice particle orientation without making the radiative transfer calculations too costly. This extension is required to make full use of ICI's double polarisation channels at 243 and 664 GHz.

In this paper we have tried to reflect the efforts already performed to prepare for inversions using ICI, but also to indicate the work that remains to be done. As a last remark we would like to stress that ICI will provide the first "operational" observations
of our atmosphere in the sub-millimetre region and its data will cover more than 20 years. This will give the weather forecasting and climate communities a new important data source.

*Author contributions.* PE and BE participated in most of the activities described and lead the manuscript writing. VM and CA contributed to the work, supervising and reviewing the algorithm development in its various phases and contributed to the manuscript writing. AT is in charge of the algorithm development inside NWC-SAF and has revised the manuscript. UK is main ICI scientist at ESA/ESTEC and has
provided the techical description. SB has contributed input and text, and revised the manuscript.

*Code availability.* The footprint remapping toolbox can be obtained by contacting EUMETSAT. It will be distributed "as is" with no warranties and on the condition of no redistribution, as well as that this article and the EUMETSAT study with Contract EUM/C0/18/4600002075/CJA are cited where used. Other numerical results in the article are based on various Matlab and Python scripts, kept in local SVN repositories, that can be obtained by contacting the first two authors. Usage of these scripts requires assess to the ARTS and Atmlab packages, available
at www.radiativetransfer.org.

*Competing interests.* The authors declare that they have no conflict of interest.

*Acknowledgements.* A first version of the algorithm, based on neural nets, was outlined by Gerrit Holl as project scientist at SMHI. The development of footprint remapping routines for ICI was performed under the EUMETSAT study " Application of optimal interpolation procedures to EPS-SG MWI and ICI ", Contract EUM/C0/18/4600002075/CJA. Figure 8 was produced by Simon Pfreundschuh (Chalmers).
The science advisory group around MWI and ICI has given feedback on the work described here. This project would not have been possible without the many individuals that are contributing to the technical development of ICI and the development of the ARTS infrastructure. The preparations for ICI performed at Chalmers University of Technology are largely funded by grants from the Swedish National Space





Agency (SNSA). The contribution of SB was supported by the Deutsche Forschungsgemeinschaft (DFG, German Research Foundation) under Germany's Excellence Strategy — EXC 2037 'Climate, Climatic Change, and Society' — Project Number: 390683824.



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





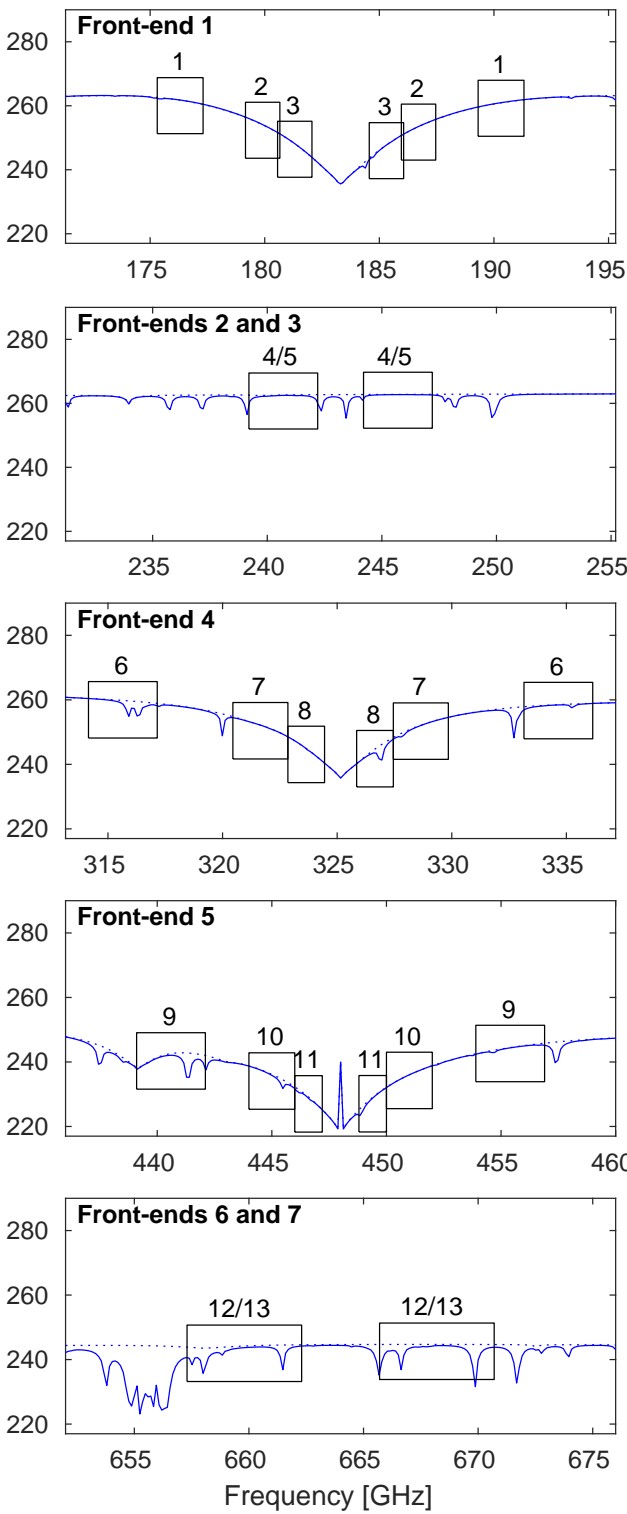

**Figure 1.** Frequency coverage of the sidebands for each ICI channel. The simulated spectrum (blue line) is based on a mid-latitude winter scenario. The dotted lines are simulations ignoring ozone.





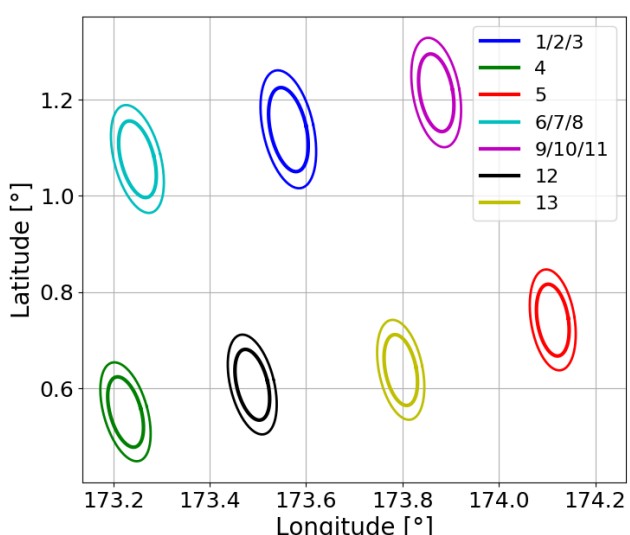

**Figure 2.** Instantaneous ICI footprints. The inner and outer contours represent the -3 and -6 dB level of normalised antenna patterns. The assumed sensor position is $6.9°$S, $175.3°$E at an altitude of $824.5$ km.

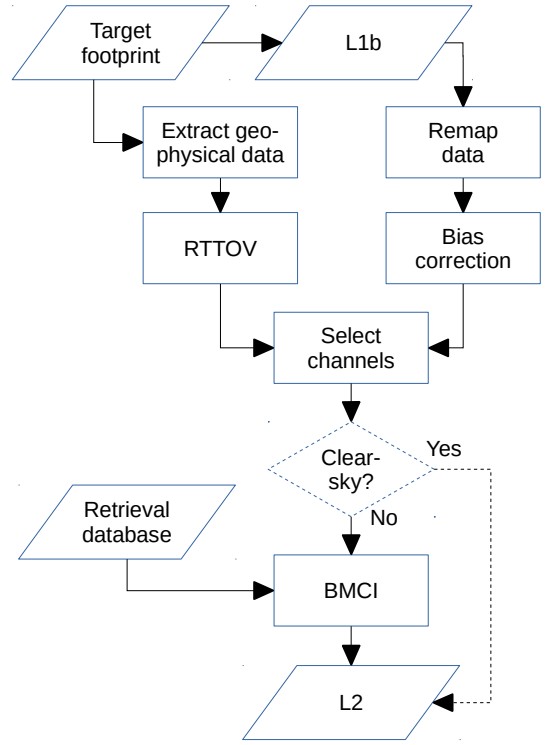

**Figure 3.** The overall data flow of the algorithm.



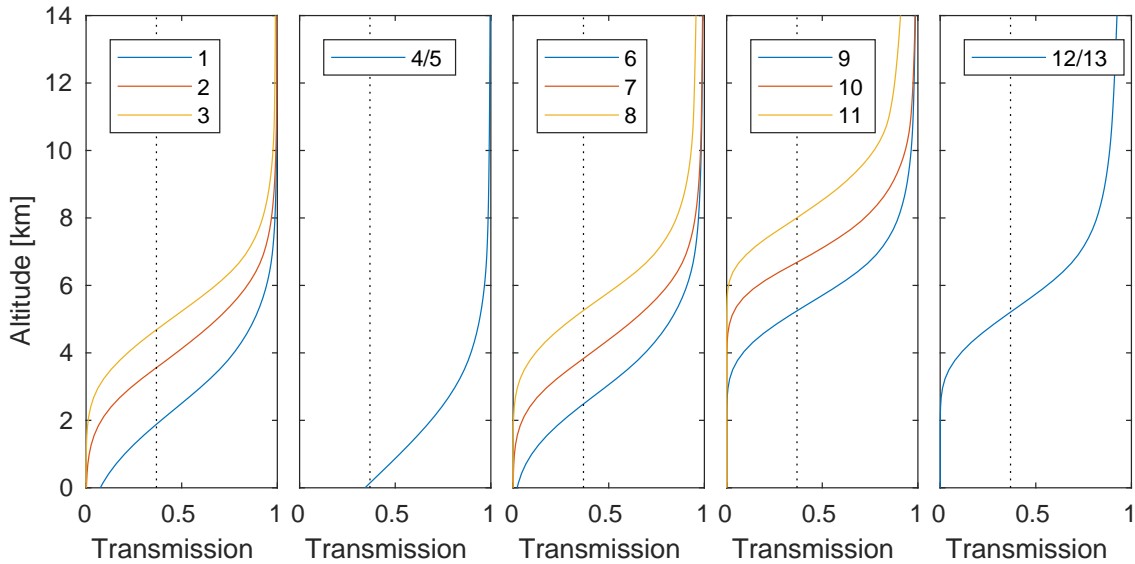

**Figure 4.** Channel mean transmission between altitudes in the atmosphere and ICI, according to a mid-latitude winter scenario. The dotted line corresponds to an optical thickness of 1.

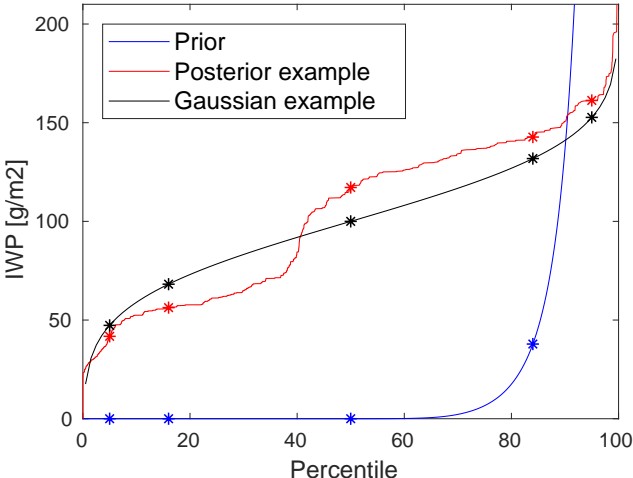

**Figure 5.** Example quantile functions. The blue line represents the retrieval database applied in Sec. 4.3, acting as prior for a test retrieval (red line). For example, the prior and posterior median values are 0 and $117\,\mathrm{g/m^2}$, respectively. The black line matches a hypothetical retrieval having a Gaussian posterior of $100 \pm 32\,\mathrm{g/m^2}$. The symbol * identifies the 5th, 16th, 50th, 84th and 95th percentiles of each distribution.





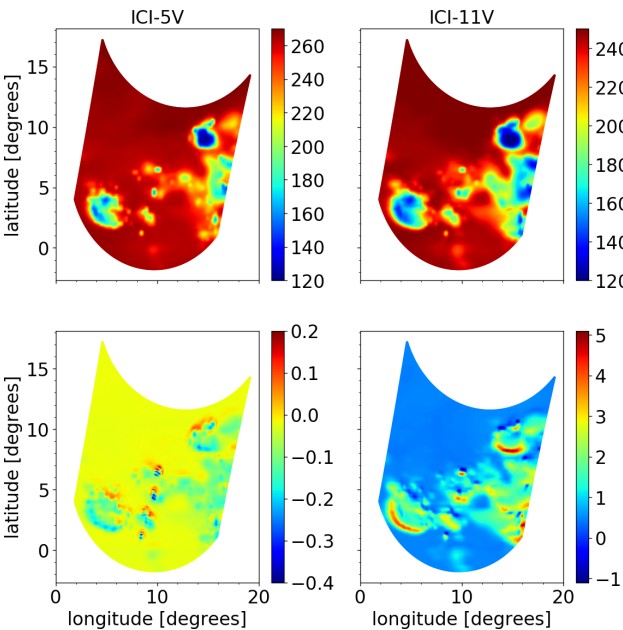

**Figure 6.** Remapping of ICI-5V and ICI-11V for an example scene. The upper row shows simulations representing the expected result after remapping (figures showing the data before remapping look identical plotted in this manner). The lower row displays the error found when remapping simulated, noise-free, observations.

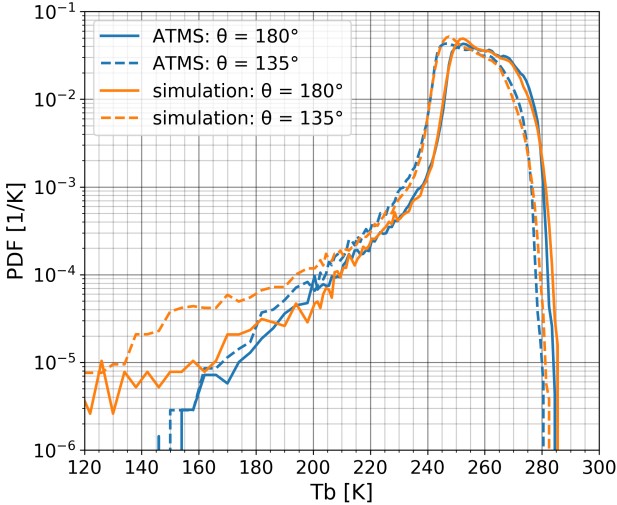

**Figure 7.** Statistical comparison of simulated and real ATMS channel 21 (183.31±1.8 GHz) measurements, for both a zenith angle of 180° (nadir) and 135° (roughly the one of ICI) and data collected between 15°S to 15°N August 2015.





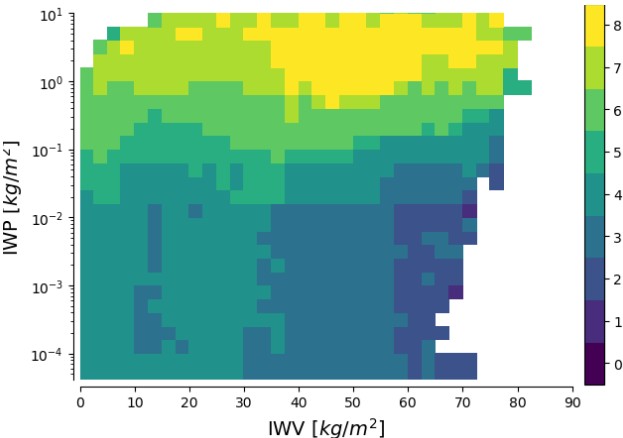

**Figure 8.** Estimated degrees of freedom (DOF) of ICI observations, as a function of integrated water vapour (IWV) and IWP. Based on the tropical part of the retrieval database (Sec. 4.3.1). Calculated by a singular value decomposition and considering the NE$\Delta$T reported in Table 1.



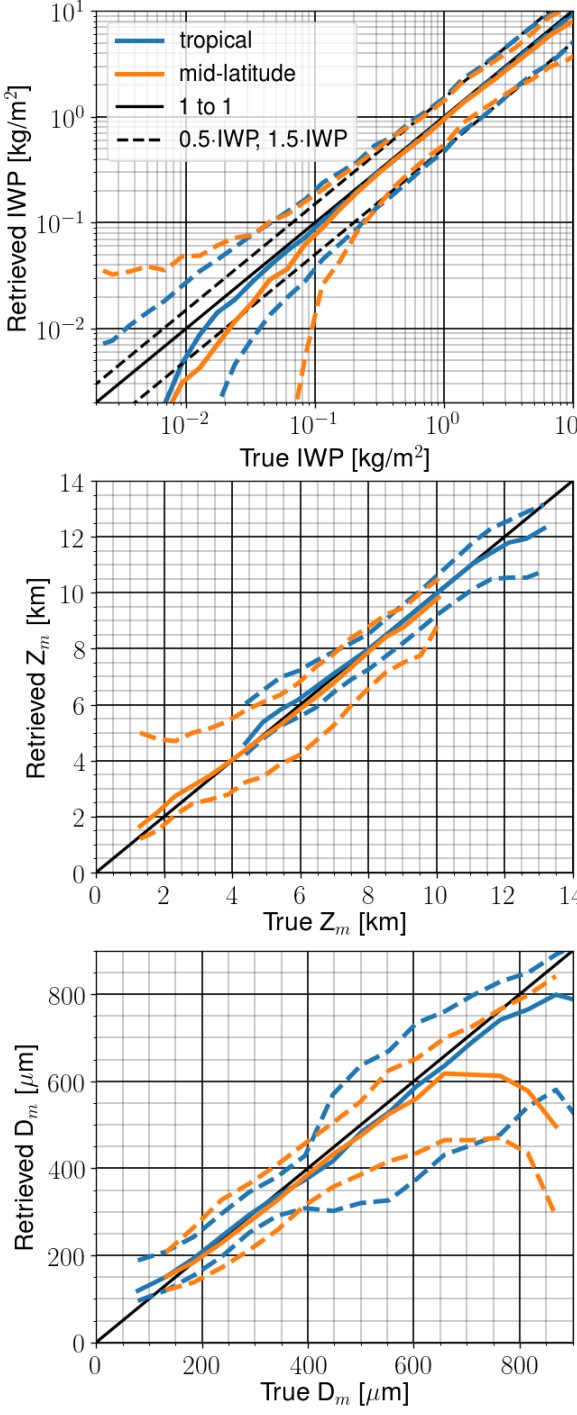

**Figure 9.** Estimated retrieval performance for IWP (top panel), $Z_m$ (Eq. 4, middle panel) and $D_m$ (Eq. 5, bottom panel). Tropical refers to data at latitudes between $30°$S and $30°$N, while mid-latitude includes data for January and February between $35°-65°$N. The blue and yellow solid lines show the median of retrieved median value, while the corresponding dashed lines show the median of retrieved 5th and 95th percentile.