# Peer review of "Towards an operational Ice Cloud Imager (ICI) retrieval product"

_Atmospheric Measurement Techniques, 2019_

## Referee Comment (RC1) · Ralph Ferraro (Referee) · 24 Sep 2019

This is an extremely well written and important paper that presents a detailed description of the upcoming ICI sensor and the baseline retrieval algorithm for a set of level 2 ice products. It will serve as the benchmark paper for a brand new sensor that will fly on the EUMETSAT second generation of satellites to be launched in the next 3-4 years.

I have no serious technical questions in the paper, aside from perhaps the consideration of using retrievals from the companion sensors such as MWI and MWS in the algorithm after launch. Perhaps this is out of scope for this initial paper but feel its worth mentioning.

Some minor comments include: 1. P1, L7 - change - "This is to prefers for ..." to "This is preferred..." 2. P2 - Para. 2 - Would it be worthwhile to define this history of sensors as "imagers" and "sounders"? 3. P2 - L10 - Change "...region upwards..." to "region to higher frequencies." 4. P2 - L17 - Change "receivers" to "sounders" 5. P2 - L21 - Change "...dealing with..." to "using" 6. P4 - L4 - Change "...includes also..." to "also includes" 7. P11 - L15 - Change "Wind speed is effectively...." to "Wind speed is applicable only over water."

---

## Referee Comment (RC2) · Anonymous Referee #1 · 25 Sep 2019

General Comments:

This aptly titled manuscript describes the considerable progress towards a "day one" retrieval algorithm for the Ice Cloud Imager on the next generation of EUMETSAT polar orbiters. As such it is highly relevant for publication in AMT. The algorithm described is an marked advance over other operational satellite retrieval algorithms in that it does not assume Gaussian statistics and retrieves five quantiles (e.g. median, 5%, and 95%) of the posterior probability distributions. The introduction provides a brief history of microwave remote sensing in general and a comprehensive summary of previous work in sub-millemetre remote sensing of ice hydrometeors. The limitations of the current algorithm and future work required are frankly and appropriately described, especially in section 5. The manuscript is well organized and the presentation is clear

(though some grammar needs to be polished).

As is necessary in this situation, the manuscript is somewhat of a summary of a detailed technical report. The performance test section, however, does need to provide information about the numerical experiments that produced the results described. Since the manuscript describes an existing algorithm (even though preliminary), it is not appropriate for a reviewer to tell the authors to make different choices about aspects of the algorithm. Instead the reviewer can help to make the presentation clear and complete, and also make suggestions which might influence future algorithm changes.

Specific Comments:

1. Section 3.4.4: Why are channels masked out if they have any surface contribution? Equation 13 shows the Bayesian uncertainty is increased for channels with surface contribution, and in a Bayesian context that approach is all that should be needed to deal with the uncertainty of surface emissivity. Won't the varying number of channels included in the retrievals cause spurious (if not statistically significant) discontinuities in the retrievals between pixels with close brightness temperatures? Perhaps some more justifications or caveats are needed in this discussion.

2. Section 3.5.1: The quantile retrieval approach requires a substantial number of database cases that match the observations so that the posterior probabilities are high enough. How is this assured? How can we trust the retrieved quantiles, especially at 5% and 95%?

3. Section 3.5.3: Could you mention how the database extraction filtering does not exclude cases which would contribute significantly to the integral and thus bias the result?

4. The end of section 3.5.3 discusses the important issue of very few or no database cases matching an observation.

4a) Do you have a non-arbitrary method for determining when there are too few matching database cases or too wide a range of probabilities for the matching database cases? What specifically are the criteria for increasing the variances in S_0?

4b) This problem is exacerbated by the Gaussian pdf assumption for the probability of the difference between an observation and the database simulation. Could you consider using a long-tailed probability distribution, which would be justified by systematic errors, such as various modeling errors?

5. Section 4.1 (Remapping of data) needs to summarize the numerical experiment performed in addition to discussing the results.

6. Section 4.1: Why is the incidence angle relevant for remapping errors for a homogeneous scene? The brightness temperatures simulated in the database can use the correct zenith angle for each channel, right?

7. Section 4.2 (Generation of retrieval database): It would be useful to include some details about the method for generating the retrieval database used in the experiments in this article.

8. Section 4.3.1 (Test retrieval database): How many CloudSat profiles were used? Do the CloudSat profiles correspond to the same 15S to 15N region and the same time (August 2015)?

9. Section 4.3.2 (Degrees of freedom): Again, there needs to be some explanation of the method used here. How is DOF calculated? The bit of explation in the short figure caption is not enough.

10. Section 4.3.2: 448.0+-1.4 GHz is an upper troposphere water vapor channel in the Tropics and is considerably more sensitive than the 183 and 325 GHz channels, so one doesn't want to give the impression that the three channels for each water vapor absorption line are equivalent. Is the DOF for low IWP and IWV 3 or 4 (I'm having trouble telling from the colour scale)?

11. Section 4.3.3 (Overall performance): A description of the method needed. Do the

two regions/seasons (tropical and mid-latitude) use different retrieval databases? Is there a minimum retrieved IWP for including cases in the Zm and Dm retrieval performance graphs?

12. The brief section 4.3.4 (Test inversions) should be omitted. It references inversion tests with ISMAR data, but without any validation or results, and thus is not very meaningful.

13. Section 5 (Outlook): Do you have ideas for how to include particle orientation in the algorithm? If not too speculative, your ideas would be interesting in this section.

14. Section 5: Another important extension to mention is including a wider range of particle size distribution variations. Presumably the variations in Dm vs IWC curves between single beams is larger than between published climatologies. Also the width of ice particle size distributions is important for relating CloudSat radar reflectivity to IWC for the prior probabilities. This issue might lead to a significant underestimate in the retrieval errors.

Technical Corrections:

Author contributions: I think "PE and BE" should be "PE and BR".

---

## Referee Comment (RC3) · Patrick Eriksson et al. · 7 Oct 2019

Review on: Towards an operational Ice Cloud Imager (ICI) retrieval product

This paper describe an operational algorithm to retrieve ice cloud properties from the Ice Cloud Imager (ICI) instrument which will be part of the second generation of the EUMETSAT Polar System (EPS-SG). The retrieval algorithm is based on a Bayesian Monte Carlo approach and allows to retrieve three main quantities, the ice water path (IWP), the mean mass diameter (Dm) and the mean mass height (Zm). The main novelty of this algorithm comes from the fact that the posterior probability function is directly Ânˇ'¡ăretrievedˇăⁱ'¡ż (do not assume any shape like gaussian) and is used to compute the ice cloud properties. This algorithm is also tested via a retrieval database in order to test the algorithm performance. Some estimation of the errors due to remapping of the 13 channels of ICI are also presented.

General comments: The paper as a whole is rather well written. However sometime the author wants to go too fast and forget the basics which are a minimum of explanation (or reference) on the methods used to infer some scientific conclusion. This is especially true in the section about the Degree of Freedom. In the later there are absolutely no explanation on how to compute this Ấńdegree of freedomẬż, no explanation on the assumption made, and no reference that could help to understand the physical meaning of this quantity. Authors often forget that potential readers of their work are not necessarily specialist in the micro-wave and ice cloud field, which in turn make sometime the paper not easy to understand or not easy to find related paper that would help our understanding. In the same idea there is often a lake of explanation on how/why an equation or a given form to simulate a physical process was chosen. For example there is no explanation on why the weigh (wi) used to compute the posterior probability was computed as in equation (9). Same concerning equation (13), we do not understand why and how the author choose to use this specific form to compute error due to a miss knowledge of the surface emissivity, or scattering within the cloud. There is also some confusion between probability and probability density function (eq 10 and 11).

So for all this reasons I think this interesting paper can be largely improved and therefore my overall recommendation is MAJOR REVISION.

Below are listed my comments related to each section:

2 - Ice Cloud Imager 2.2 The receiver package

line 19: I am not in the measurements field and have hard time to understand what is the receiver noise temperature and especially the very large values of 600 to 2600K. Maybe a reference could help the reader to find information on this receiver noise temperature.

3 - Algorithm 3.2 Overview

line 29: It is actually not evident to get this document, I tried the link ÂńÂădocumenta-tionÂăÂż under www.nwcsaf.org, but did not find any document related to ICI. Anyway, maybe some annexe would help the reader to understand the details of the algorithm or an academic reference.

3.3 Input and Output

line 10: What is the dimension (unit) of r? Specify it in the text. You could also be explicit on the solid angle by saying that the solid angle is the one of the antenna (I guess). Concerning the cartesian coordinate we have no idea on where is the origin of this cartesian system. Please specify also if the antenna pattern is taken on the ground, or change with altitude and therefore you need to know the cloud extent (information that you do not have). It is not so evident how you concretely compute IWP.

line 12: dveq is the equivalent volume diameter but equivalent to what? Spherical particles? Specify it in the text.

line 17: the Author call Dm the mean mass size but Delanoe et al. 2014 expressed it as the volume-weighted diameter because it is the 4th moment of the size distribution over the third one. Why did you call it mean mass size? We don't see any mass weighted in the formulae!

3.4 Pre-processing part 3.4.1 Target footprint and rem apping of data

line 9: what do you mean by semi-circular?

3.4.5 Detection of clear sky data

line 23: Last sentence ÂńAs the module likely will not be applied, no details are here givenÂăÂż. So if you do not give any detail about the clear sky module detection of the algorithm, do we really need this paragraph?

3.5 Inversion part 3.5.1 Theory and retrieval representation

line 4 (eq. 9): Could you justify why wi is written like an exponential law?

line 5: Why only observation uncertainties are taken into account in S0? Can't you add the forward model uncertainties also (due to the miss-knowledge of non retrieve parameters that play in the forward model to compute yi)?

line 6: The following sentence is very hard to understand, please rephrase it. ÂńThey are not standard, but are introduced to allow tailoring of the retrieval database to the specifics of the retrievals of concern.ÂăÂż

line 14 and 16 (eq. 10 and 11): Something is wrong here because p(x'|y) is what we call a probability density fuction (PDF) in (10), we need to multiply by dx to get a probability. But in (11) p(xi|y) should be a probability but the notation is almost identical to the previous equation (10), and make this 2 equation confusing. Specify somewhere that p(xi|y) is not a PDF but directly a probability to have x=xi.

3.5.2 Measurement vector and uncertainties

line 2: Why is there only one clear sky in your database? Don't you need to simulate different clear sky at least to take into account the different emissivity, surface temperature that depend of the season and location?

line 7 (and eq. 13): We don't understand why the uncertainty on emissivity take this form, could you explain a bit or give some reference? We don't even know what is tau_e,j?

line 9 (and eq. 13): I really don't understand why this last term of equation 13, in its present form, could model the uncertainty due to a miss-representation of the scattering in the model! Please explain why in the text or give a reference!

line 10: Why can you assume that the modeling error (scattering) is proportional to the deviation of clear-sky reference simulation? Where does this assumption comes from? Any Reference?

3.5.3 Database extraction and iterations: OK now I understand a bit more why there is only one clear sky in the database! You should put this paragraph before the paragraph

3.5.2!

4 Performance tests 4.1 Remapping of data

line 8: On figure 6 please indicate the units of the color scale

line 18-19: Difficult to understand the following sentence, maybe a figure could help here! ÂńÂăThis means that line-of-sights of observations and the corresponding ones after remapping cross at the ellipsoid but deviate at altitudes inside the atmosphere.ÂăÂż

4.3 ICI retrieval performance 4.3.1 Test retrieval database

line 19:what do you mean by unrealistically high? Give some number. (Correct the word spelling also)

line 22: why around 0.8? Don't you have an exact number or did you make some random choice around 0.8?

line 24-25: This comparison between ATMS observations and simulations are done over which period? Does it used every observation or is there some filtering? Are you taking into account any specific antenna pattern of the ATMS instrument? How the atmospheric profile and hydrometeors are define? Are you using Cloudsat also? Are you using the same definition for the microphysical model than for ICI? Please develop in order to help the reader to understand the limit of this statistical comparison!

line 29: Please explain how the particle orientation can explain this discrepancy?

line 31: What is GMI? Any reference?

Table 2: Could you please indicate the habit model explicitly instead of a number referring to another paper from the author.

4.3.2 Degrees of freedom

line 7-8: The degree of freedom is more commonly called DoFS, Degree of Freedom

for Signal (See Rodgers, 2000), and it can be seen as Âń Âăthe number of independent peace of information given by the observing systemÂă(its maximum value is the size of the state vector)Âż which is not exactly Âń the effective number of channelsÂăÂż

line 9-10: Give some information on how you compute this degree of freedom for signal! At least a reference! To compute the DoFS you need to specify a state vector, could the author give this state vector here? It seems that now it contains the humidity profile, and not only the IWP, Zm and Dm? The value of the DoFS is also highly dependent of the a-priori definition and especially the prior variance-covariance matrix. Please develop!

line 13: Which surface parameter are you talking about, emissivity or surface temperature?

line 21-22: This is why we also compute the posterior error together with the DoFS!

4.3.3 Overall performance

line 6: I may have missed it somewhere but what is H?

line 8 (Fig 9): The problem with using only average is that we have no idea of the dispersion around the mean. A scatter plot presented with a 2D colorscale histogram give much more information on the overall performance of the retrieval. The author should consider this king of plot instead of presenting only the average.

line 9: What do you mean by "good accuracy"? Is it in comparison to other related retrieval from other sensor?

line 14: Are the 5th and 95th percentile also averaged values?

line 17: This precision number for Zm and Dm are average precision, specify it somewhere!

4.3.4 Test inversions

line 4: Reference on ISMAR?

line 8-9: last sentence says that ÂńÂăThese various results have been presented at conferences (e.g. Eriksson et al., 2016) and the details are not repeated here.ÂăÂż If I follow your thought, what the point to write this paper if you already presented at conference (e.g. Ericksson et al. 2016)? I still think that an academic referenced paper is deeper than a conference paper and it should therefore be the way around, the conference paper at some point should cite this paper. . .

5 Outlook

line 12-15: The cloudsat and caliop based algorithm like DARDAR for example, which retrieve IWC profile from the combinaison of both measurements, often show a layer of supercooled water above the ice layer in the polar area. Are you planing to integrate this kind of case in your database in the futur. Is microwave sensitive to this kind of situation?

---

## Short Comment (SC1) · 17 Oct 2019

The authors present a very useful overview of the ICI instrument and the day-1 retrieval algorithm. However, I feel that the results in Sec 4.3.3, particularly for the midlatitude case, may rather over-estimate the retrieval accuracy due to the use of a single ice particle habit and size distribution. I would like to see some evaluation of the impact of this assumption on the retrieval since previous work e.g. Fox et al., AMT, 2019 (amt-12-1599-2019) have shown a strong sensitivity of sub-mm brightness temperature to particle habit. Presumably if multiple habits and PSDs are included in the database (as for the tropical case) this will increase the uncertainty in the retrieval?

I also find the description of the radiative transfer simulations in Sec 4.3.1 a little con-

fusing, particularly how the 2D CloudSat information is used. As far as I am aware the ARTS RT4 solver can only handle 1-dimensional atmospheres. The discussion in Sec 5 suggests that an independent beam approximation is used to handle the 2D antenna pattern, but this is not fully described.

───────────────────────────────

---

## Author Response (AR1)

Dear Ralph Ferraro,

Thanks for the kind words about our manuscript.

Joint MWI-ICI retrievals are already mentioned in Sec. 5, but we will add a comment on this possibility in Conclusions. We avoid discussing MWS as it will be on the other platform and the time difference between platforms A and B will be relatively high, about 50 min (SG-1B will be positioned at about 180 degrees relative to SG-1A).

All your minor comments will be followed.

Best regards,

Patrick and co-authors

[Figure]

Yes, it is unlucky that we have only a single habit for mid-latitudes and, as you point out, that may give an over-estimation of the retrieval performance. We also write that the mid-latitude results shall be approached with care (page 15, lines 20-21). We will add a clarification that the lack of multiple habits is one reason for the concern. We will also stress this stronger in the Outlook part.

We would like to mention that the efforts so far have focused on the core algorithm and the retrieval database discussed has been produced as an initial working basis. Future

studies will be required to expand the database. Accordingly, it could be produced by persons not involved in the work presented here. We will clarify in the revised manuscript.

We will better explain that RT4 indeed is applied following the independent beam approximation.

Best regards,

Patrick and co-authors
* * *
Atmos. Meas. Tech. Discuss.,
doi:10.5194/amt-2019-312-AC3, 2019

[Figure]

Below we will try to answer your questions and what changes of the manuscript we will implement. Your comments are in italic, with our answers below.

As the retrieval database is in the background of many of the questions, we start by mentioning that the efforts so far have focused on the core algorithm and the retrieval database discussed has been produced has to be considered an initial working basis. Future studies will be required to expand the database.

*1. Section 3.4.4: Why are channels masked out if they have any surface contribution? Equation 13 shows the Bayesian uncertainty is increased for channels with surface*

*contribution, and in a Bayesian context that approach is all that should be needed to deal with the uncertainty of surface emissivity. Won't the varying number of channels included in the retrievals cause spurious (if not statistically significant) discontinuities in the retrievals between pixels with close brightness temperatures? Perhaps some more justifications or caveats are needed in this discussion.*

It is correct that the surface term in Eq. 13 and the channel selection to some extent do the same thing. However, Eq. 13 models the uncertainty in a simplistic manner. The main limitation is that the correlation between channels is ignored. An error in assumed surface emissivity will cause an error that is correlated between channels, but presently we have no knowledge at all on this correlation at sub-mm wavelengths. There should be correlation, but it is likely not one between all channels. See further answer to question 4b.

In any case, if found unnecessary the channel selection can be "deactivated" by setting the optical thickness thresholds to some very high value.

We will add comments to make these aspects clearer.

Yes, there is a high risk that there will be discontinuities in the retrieval at e.g. coasts. They will to a large extent dependent on the quality of the final retrieval database. It could be mentioned that many microwave products are ocean-only, due to the problem of modelling land emissivities, but we wanted to at least keep the door open for producing results over land.

*2. Section 3.5.1: The quantile retrieval approach requires a substantial number of database cases that match the observations so that the posterior probabilities are high enough. How is this assured? How can we trust the retrieved quantiles, especially at 5% and 95%?*

To our best knowledge, there is no manner to set a required number. The general rule is simple: the more the better! This means that the final retrieval database shall

represent as many cases as possible, and to make this easier we have included the possibility to use a prior weights ($a_i$).

At the end of the outlook section, we make a comment about that machine learning seems to decrease the demand on database size, and that to apply BMCI for combined MWI and ICI retrievals are probably out of the question.

*3. Section 3.5.3: Could you mention how the database extraction filtering does not exclude cases which would contribute significantly to the integral and thus bias the result?*

Nothing like this is built into the algorithm. It will depend on sensible choices of the configuration parameters. The recommendation is, of course, to use broad ranges for inclusion and that tests should be performed when the final retrieval database is at hand.

*4. The end of section 3.5.3 discusses the important issue of very few or no database cases matching an observation. 4a) Do you have a non-arbitrary method for determining when there are too few matching database cases or too wide a range of probabilities for the matching database cases? What specifically are the criteria for increasing the variances in $S_0$?*

There is no automatic method to set the required number of matches. The number shall be set as high as possible considering the size of the final retrieval database.

We don't see a problem if there is a wide a range of probabilities for the matching database cases. On the contrary, if the range is narrow then you have likely not sampled a sufficient broad part of the a priori distribution.

The rules for increasing variances are important, but require quite some space to describe. For this reason, we decided to only refer to Rydberg (2018) regarding this part.

*4b) This problem is exacerbated by the Gaussian pdf assumption for the probability of the difference between an observation and the database simulation. Could you con-*

*sider using a long-tailed probability distribution, which would be justified by systematic errors, such as various modelling errors?*

BMCI is more flexible here than OEM (1D-VAR), but BMCI stills assumes that all uncertainties that are covered by $S_o$ approximately follow Gaussian statistics. So on that side the answer is no. On the other hand, in the generation of the database you are not limited to Gaussian assumptions. (We will add text to clarify this) This means that "outlier cases" can be included in the database.

Above we discussed modelling of surface emissivity. In the generation of the database the emissivity can be varied, and if the knowledge is poor the surface emissivity should be given a broad distribution (that does not need to be Gaussian). This is relatively straightforward. The problem is that the correlation between frequencies of emissivity variations also has an impact, and here reasonable assumptions are very hard to formulate before we actually have the ICI data.

*5. Section 4.1 (Remapping of data) needs to summarise the numerical experiment performed in addition to discussing the results.*

Will be done.

*6. Section 4.1: Why is the incidence angle relevant for remapping errors for a homogeneous scene? The brightness temperatures simulated in the database can use the correct zenith angle for each channel, right?*

We briefly discussed that in the Outlook section. We will add some text to Sec 4.1 to make this more clear.

*7. Section 4.2 (Generation of retrieval database): It would be useful to include some details about the method for generating the retrieval database used in the experiments in this article.*

We see Sec. 4.2 as a general discussion. Details of generation of the database used are found in Sec 4.3.1 (that will be somewhat expanded).

8. Section 4.3.1 (Test retrieval database): How many CloudSat profiles were used? Do the CloudSat profiles correspond to the same 15S to 15N region and the same time(August 2015)?

A smaller dataset, compared to the retrieval database, is used in Fig. 7. We will make this clear.

*9. Section 4.3.2 (Degrees of freedom): Again, there needs to be some explanation of the method used here. How is DOF calculated? The bit of explanation in the short figure caption is not enough.*

A description of how DOF is calculated will be added.

*10. Section 4.3.2: 448.0+-1.4 GHz is an upper troposphere water vapour channel in the Tropics and is considerably more sensitive than the 183 and 325 GHz channels, so one doesn't want to give the impression that the three channels for each water vapor absorption line are equivalent. Is the DOF for low IWP and IWV 3 or 4 (I'm having trouble telling from the colour scale)?*

It is correct that the 448 GHz transition is stronger than the 183 and 325 GHz ones, and has the potential to provide information at higher altitudes than the two later transitions. Figure 1 supports this. However, the present NEDT values for the 448 GHz front-end are relatively high. We have not performed a dedicated test, but we suspect that the assumed NEDT gives too noisy data for obtaining information from the two innermost 448 GHz channels, for single-footprint retrievals. (NEDT is also high for 325 GHz, and likely 183 GHz dominates the IWV information content in Fig 8.) We will make a small rewording, and leaves the details to a dedicated study on ICI's performance at clear-sky conditions.

The DOF for low IWP and IWV is 4. We will redo the figure with another colour scale.

*11. Section 4.3.3 (Overall performance): A description of the method needed. Do the two regions/seasons (tropical and mid-latitude) use different retrieval databases? Is*

*there a minimum retrieved IWP for including cases in the Zm and Dm retrieval performance graphs?*

There is just one single retrieval database. We will clarify how the database is used. Yes, an IWP threshold is applied in Fig. 9. Thanks for spotting this. We will add this information.

*12. The brief section 4.3.4 (Test inversions) should be omitted. It references inversion tests with ISMAR data, but without any validation or results, and thus is not very meaningful.*

We will follow this advice.

*13. Section 5 (Outlook): Do you have ideas for how to include particle orientation in the algorithm? If not too speculative, your ideas would be interesting in this section.*

In fact, we have been working on this topic, and the first data and results just appeared in AMTD. So we will gladly write a few words about this.

*14. Section 5: Another important extension to mention is including a wider range of particle size distribution variations. Presumably, the variations in Dm vs IWC curves between single beams is larger than between published climatologies. Also, the width of ice particle size distributions is important for relating CloudSat radar reflectivity to IWC for the prior probabilities. This issue might lead to a significant underestimate in the retrieval errors.*

Yes, totally correct. The present text does not make this totally clear. We will rewrite.

Best regards,

Patrick and co-authors

Atmos. Meas. Tech. Discuss.,
doi:10.5194/amt-2019-312-AC4, 2019

[Figure]
We deliberately tried to keep the manuscript as short as possible, to not let details cause distraction, but we have clearly been too brief in some parts. The other reviewers made similar remarks for some parts. We will expand the manuscript with the aim to make it more easily accessible for a broader audience.

Below we comment on your specific questions and what changes of the manuscript we will implement. Your comments are in italic, with our answers below.

*line 19: I am not in the measurements field and have hard time to understand what is the receiver noise temperature and especially the very large values of 600 to 2600K. Maybe a reference could help the reader to find information on this receiver noise*

*temperature.*

We will add a reference for receiver noise temperature. Much lower values can be achieved by cooled receivers, but cryogenic cooling represents a risk and a relatively short life-time. Hence, such techniques can not be used for operational missions.

*line 29: It is actually not evident to get this document, I tried the link documentation under www.nwcsaf.org, but did not find any document related to ICI. Anyway, maybe some annexe would help the reader to understand the details of the algorithm or an academic reference.*

We are deeply sorry, but the information in the reference is wrong. The mistake will be corrected.

*line 10: What is the dimension (unit) of r? Specify it in the text. You could also be explicit on the solid angle by saying that the solid angle is the one of the antenna (I guess). Concerning the cartesian coordinate we have no idea on where is the origin of this cartesian system. Please specify also if the antenna pattern is taken on the ground, or change with altitude and therefore you need to know the cloud extent (information that you do not have). It is not so evident how you concretely compute IWP.*

We will clarify.

*line 12: dveq is the equivalent volume diameter but equivalent to what? Spherical particles? Specify it in the text.*

Will be done.

*line 17: the Author call Dm the mean mass size but Delanoe et al. 2014 expressed it as the volume-weighted diameter because it is the 4th moment of the size distribution over the third one. Why did you call it mean mass size? We don't see any mass weighted in the formulae!*

As mass is the volume times density, the volume-weighted and mass-weighted are

equal concepts. Or consider equation 3, showing that $d_{veq}^3$ is directly proportional to mass. This means that $d_{veq}^3$ in both integrals represent the mass (factors such as density and $\pi$ cancel out).

*line 9: what do you mean by semi-circular?*

We mean "close to circular". We will rephrase.

*line 23: Last sentence "As the module likely will not be applied, no details are here given". So if you do not give any detail about the clear sky module detection of the algorithm, do we really need this paragraph?*

It is described in Rydberg, 2019. We will make this clear.

*line 4 (eq. 9): Could you justify why wi is written like an exponential law?*

We will add a reference.

*line 5: Why only observation uncertainties are taken into account in S0? Can't you add the forward model uncertainties also (due to the miss-knowledge of non retrieve parameters that play in the forward model to compute yi)?*

We used the term "observation uncertainty" as the uncertainty of the observation system, thus including the forward model. We will rephrase to be clearer.

*line 6: The following sentence is very hard to understand, please rephrase it. "They are not standard, but are introduced to allow tailoring of the retrieval database to the specifics of the retrievals of concern."*

Will be done.

*line 14 and 16 (eq. 10 and 11): Something is wrong here because $p(x'|y)$ is what we call a probability density function (PDF) in (10), we need to multiply by dx to get a probability. But in (11) $p(xi|y)$ should be a probability but the notation is almost identical to the previous equation (10), and make this 2 equation confusing. Specify somewhere*

*that $p(xi|y)$ is not a PDF but directly a probability to have $x = xi$.*

Correct. Thanks for pointing this out. Will be fixed. (In fact, above Eq 11 it says $p_n$, that was meant to flag that it is a probability, but we missed to complete this). Eq 8 will also be corrected.

*line 2: Why is there only one clear sky in your database? Don't you need to simulate different clear sky at least to take into account the different emissivity, surface temperature that depend of the season and location?*

On the database side, the assumption is that we know the true values and there is only one clear-sky simulation to be done. However, it is critical that the database contains cases covering a distribution of emissivities, surface temperatures etc. See further comments in the reply to referee 1. We think the comments we will add in response to referee 1 will make all this clear.

*line 7 (and eq. 13): We don't understand why the uncertainty on emissivity take this form, could you explain a bit or give some reference? We don't even know what is $\tau_{e,j}$?*

$\tau_{e,j}$ should have been $\tau_{cs,j}$. Sorry, we missed to change this equation when changing nomenclature for expressing optical thicknesses. We will explain the expression for emissivity uncertainty.

*line 9 (and eq. 13): I really don't understand why this last term of equation 13, in its present form, could model the uncertainty due to a miss-representation of the scattering in the model! Please explain why in the text or give a reference!*

See answer to the next question.

*line 10: Why can you assume that the modeling error (scattering) is proportional to the deviation of clear-sky reference simulation? Where does this assumption comes from? Any Reference?*

As we understand it, these two questions refer to the same issue. We will add an

explanation.

*3.5.3 Database extraction and iterations: OK now I understand a bit more why there is only one clear sky in the database! You should put this paragraph before the paragraph 3.5.2!*

The advice will be followed.

*line 8: On figure 6 please indicate the units of the color scale*

Will be done.

*line 18-19: Difficult to understand the following sentence, maybe a figure could help here! "This means that line-of-sights of observations and the corresponding ones after remapping cross at the ellipsoid but deviate at altitudes inside the atmosphere."*

We will rephrase.

*line 19: what do you mean by unrealistically high? Give some number. (Correct the word spelling also)*

We will add some number(s) and correct spelling.

*line 22: why around 0.8? Don't you have an exact number or did you make some random choice around 0.8?*

First, 0.8 is a "typo", should be 0.9. Let's call this an "educated guess". We will clarify that there is no reference model for land emissivity is at hand. Accordingly, this value is highly uncertain, and this is why we apply high optical thickness thresholds above land (at least 3).

*line 24-25: This comparison between ATMS observations and simulations are done over which period? Does it used every observation or is there some filtering? Are you taking into account any specific antenna pattern of the ATMS instrument? How the atmospheric profile and hydrometeors are define? Are you using Cloudsat also? Are*

*you using the same definition for the microphysical model than for ICI? Please develop in order to help the reader to understand the limit of this statistical comparison!*

Except for footprint size and time period used, all is done as for the retrieval database. We will clarify this.

*line 29: Please explain how the particle orientation can explain this discrepancy?*

Will be done.

*line 31: What is GMI? Any reference?*

GMI is introduced in the Introduction, bu the acronym will be explained again and a reference will be added.

*Table 2: Could you please indicate the habit model explicitly instead of a number referring to another paper from the author.*

Will be done (but will require a two-column table).

*line 7-8: The degree of freedom is more commonly called DoFS ...*

We will change DOF to DoF.

It was clearly a big mistake to not include a description of how we calculate the DoF. Both you and referee 1 ask for it. And by your comments we notice that we opened up for misunderstandings. In short, the DoF we display matches Rodgers' section 2.4.1. We tried to indicate this by defining the DoF as "measurements' degrees of freedom". Our comments seem to be based on the assumption that our DoF is the one described in Rodgers' 2.4.2. As this is not correct we don't go into details here.

We will describe the way we calculate DoF and clarify that it matches Rodgers 2.4.1.

*line 13: Which surface parameter are you talking about, emissivity or surface temperature?*

Both, and other ones. We will rewrite to something like: the various variables affecting

surface emission and reflectivity

*line 6: I may have missed it somewhere but what is H?*

We will change to H-polarisation.

*line 8 (Fig 9): The problem with using only average is that we have no idea of the dispersion around the mean. A scatter plot presented with a 2D colorscale histogram give much more information on the overall performance of the retrieval. The author should consider this king of plot instead of presenting only the average.*

The figure is based on so many retrievals that it would impossible to discern individual cases in a scatter plot, and we don't see this an option.

*line 9: What do you mean by "good accuracy"? Is it in comparison to other related retrieval from other sensor?*

Yes, this a vague statement. Will be rephrased.

*line 14: Are the 5th and 95th percentile also averaged values?*

Median. This information is found in the figure text.

*line 17: This precision number for Zm and Dm are average precision, specify it somewhere!*

Will be done.

*line 4: Reference on ISMAR?*

ISMAR is introduced and referenced in Sec 1.

*line 8-9: last sentence says that ...*

This section will be removed, following a recommendation of referee 1.

*line 12-15: The cloudsat and caliop based algorithm like DARDAR for example, which retrieve IWC profile from the combination of both measurements, often show a layer of*

*supercooled water above the ice layer in the polar area. Are you planing to integrate this kind of case in your database in the futur. Is microwave sensitive to this kind of situation?*

This is an interesting question, but, unfortunately, is an aspect of ICI observations that has got very little attention. To our best knowledge, nobody has studied the sensitivity of ICI to supercooled water for polar conditions. Indications to supercooled water for mid-latitude conditions are found in a manuscript in review (Pfreundschuh et al., AMTD).

This means that supercooled water probably must be considered in the generation of future retrieval databases. With the AMTD manuscript at hand, we now feel that it is motivated to suggest this. However, we want to here clarify that the generation of a complete retrieval database will be the subject of future studies, and it is today not known who will produce that database.

Another form of supercooled water is the liquid drops brought to sub-zero temperatures in updraft regions. These drops can have considerably size, and when present should impact on both ICI and CloudSat observations. We now notice that we missed to comment on this, and we will add a general comment/discussion of super-cooled water.

Best regards,

Patrick and co-authors

As Eq. 9 involves $\boldsymbol{S}_o$, this has the consequence that all uncertainties covered by this covariance matrix must approximately follow Gaussian statistics (as for 1DVAR). On the other hand, BMCI allows any a priori distribution of variables (unlike 1DVAR), and e.g. "outliers" can be included in the generation of the retrieval database.

The actual solution of BMCI is the estimated posterior distribution (as for all Bayesian methods), but it is unpractical to report sets of $p$. Some more compact description is needed. If the posterior distribution follows a Gaussian distribution it suffices to report the expectation value and the width of the distribution. ICI retrievals do not fall into this category and it was decided to instead use a more general description based on the cumulative distribution function, in the continuous case defined as

$$F_{x|\boldsymbol{y}}(x) = \int\limits_{-\infty}^{x} p(x'|\boldsymbol{y})\mathrm{d}x', \tag{10}$$

where $p$ denotes a probability density function, and in the framework of BMCI is obtained by summing $p_n$ for the probability of all cases having $x_i < x$:

$$F_{x_i|\boldsymbol{y}}(x) = \sum_{x_i<x} p_i(x_i|\boldsymbol{y}). \tag{11}$$

Using Eq. 11, $F_{x|\boldsymbol{y}}$ is calculated on a wide grid of $x$-values. These data are then used to obtain the inverse distribution function, $F^{-1}$, numerically by interpolation to a set of fixed percentiles. A more descriptive name of $F^{-1}$ is the quantile function. For example, $F^{-1}(0.5)$ is the median and the 90th percentile is $F^{-1}(0.9)$. Figure 5 exemplifies prior and posterior quantile functions.

It is presently planned to report the 5th, 16th, 50th, 84th and 95th percentiles in the L2 data. If the retrieval must be condensed to a single value, the first candidate to "best estimate" should be the 50th percentile. The other percentiles can be used in different ways. For example, if the 5th percentile for IWP is $> 0$ then a correct detection of ice hydrometeors is highly probable. The 16th/84th percentile range matches $\pm 1\sigma$ for a Gaussian distribution. The true value is between the 5th and 95th percentiles with a probability of 90%, etc.

**3.5.2 Measurement vector and uncertainties**

The measurement vector ($\boldsymbol{y}$) incorporates data from channels fulfilling the optical thickness criterion of Eq. 7 as a difference:

$$\Delta T_{a,j} = T_{a,j}^c - T_{a,j}^r$$

where $T_{a,j}^c$ is defined by Eq. 6 and $T_{a,j}^r$ is a simulated antenna temperature (by RTTOV, Sec. 3.4.3). To match this, the retrieval database contains both a full (all-sky) simulation and one (clear-sky) matching $T_{a,j}^r$.

The matrix $\boldsymbol{S}_o$ (Eq. 9) shall represent both instrument and simulation uncertainties. It is kept diagonal in lack of relevant information on uncertainty correlations between channels, but also for calculation efficiency reasons. The variances $\sigma^2$ are set as

$$\sigma_j^2 = \mathrm{NE\Delta T}_j^2 + (\Delta\epsilon T_{\mathrm{skin}} e^{-\tau_e,j})^2 + (c\Delta T_{a,j})^2,$$

where NEΔT is uncertainty due to thermal noise and calibration. The second term aims at representing impact of unknown surface emissivity, where $\Delta\epsilon$ is emissivity uncertainty, $T_{\text{skin}}$ is the ECMWF surface skin temperature and it is assumed that the emissivity is relatively high (impact through reflection of down-welling radiation neglected). The last term covers uncertainty in modelling of hydrometeor scattering, where it is assumed that the modelling error is proportional to the deviation from the clear-sky reference simulation. NEΔT for each channel ($j$), $\Delta\epsilon$ for water and land, and $c$ are constants, part of the configuration data.

**3.5.2 Database extraction and iterations**

Not all database cases are included in the BMCI summation, a filtering is done based on surface type, pressure, wind speed and temperature, as well as $\Delta T_a$ (as defined below in Eq. 12). Wind speed is  applicable only over water. The database extraction is done in an iterative manner, where the filter limits are adjusted with an iteration counter, in order to fetch both the most relevant and a sufficient number of matches. The filtering does not involve latitude or season. This results in that e.g. a tropical database case can influence the inversion of a mid-latitude summer measurement, if there is a match in surface temperature etc.

An additional iteration scheme has been added around the core BMCI calculations. A first reason is to better make use of the observations in situations with significant hydrometeor contents. The optical thickness associated with hydrometeors is estimated alongside of the L2 data in each iteration. Based on this updated estimate of the total optical thickness, Eq. 7 is reevaluated for all channels. If this results in that more channels can be included, BMCI is reiterated with the new channel mask. This iteration is important as the channels sensitive to the surface in a clear-sky situation, and thus ignored in the initial iteration, are the most important ones to obtain good estimates at high IWP.

The second reason is to handle the fact that the retrieval database only provides a discrete coverage of the distribution of $\boldsymbol{y}$. If one $\boldsymbol{y}_i$ happens to agree closely with $\boldsymbol{y}$, one $w_i$ can be orders of magnitude bigger than all other $w$ and the summation in Eq. 8 will be dominated by one database case. While the median value found can be realistic, this results in an underestimation of the retrieval uncertainty. It could also be the case that no $\boldsymbol{y}_i$ gives a significant match with $\boldsymbol{y}$. Both these situations are primarily handled by increasing the variances in $\boldsymbol{S}_o$, effectively making the "search radius" larger. If this does not suffice, channels will be rejected until an acceptable number of significant weights are obtained.

For further details of the filtering and iteration schemes, see Rydberg (2018). All critical parameters are part of the configuration data.

**3.5.3 Measurement vector and uncertainties**

The measurement vector ($\boldsymbol{y}$) incorporates data from channels fulfilling the optical thickness criterion of Eq. 7 as a difference:

$$\Delta T_{a,j} = T_{a,j}^c - T_{a,j}^r \tag{12}$$

where $T_{a,j}^c$ is defined by Eq. 6 and $T_{a,j}^r$ is a simulated antenna temperature (by RTTOV, Sec. 3.4.3). To match this, the retrieval database contains the difference between a full (all-sky) simulation and one (clear-sky) matching $T_{a,j}^r$.

The matrix $\boldsymbol{S}_o$ (Eq. 9) represents both instrument and simulation uncertainties. It is kept diagonal in lack of relevant information on uncertainty correlations between channels. The knowledge regarding such correlations is especially poor for surface emissivity. The variances $\sigma^2$ are set as

$$\sigma_j^2 = \mathrm{NE\Delta T}_j^2 + (\Delta\epsilon T_{\mathrm{skin}} e^{-\tau_{\mathrm{cs},j}})^2 + (c\Delta T_{a,j})^2, \tag{13}$$

where $\mathrm{NE\Delta T}$ is uncertainty due to thermal noise and calibration. The second term aims at representing the impact of unknown surface emissivity, where $\Delta\epsilon$ is emissivity uncertainty, $T_{\mathrm{skin}}$ is the ECMWF surface skin temperature, and it is assumed that the emissivity is relatively high. The antenna temperature is then approximately $T_a = \epsilon T_{\mathrm{skin}} e^{-\tau_{\mathrm{cs},j}} + T_e(1 - e^{-\tau_{\mathrm{cs},j}})$, where $T_e$ is an effective temperature of the atmosphere, and thus $dT_a/d\epsilon \approx T_{\mathrm{skin}} e^{-\tau_{\mathrm{cs},j}}$.

The last term covers uncertainty in modelling of hydrometeor scattering. To our best knowledge, no investigation of such modelling errors has been made. The uncertainty is zero for clear-sky conditions, and it should in general increase with the strength of scattering. Based on these two simple observations, we decided to simply model the error as proportional to the deviation from the clear-sky reference simulation. $\mathrm{NE\Delta T}$ for each channel ($j$), $\Delta\epsilon$ for water and land, and $c$ are constants, part of the configuration data.

**4 Performance tests**

**4.1 Remapping of data**

Samples from all ICI channels will be convolved into the field of view of ICI-1V. This section summarises the main findings obtained by applying the Backus-Gilbert toolbox developed (Sec. 3.4.1).

**4.1.1 Simulate test data**

To test the toolbox four full orbits were simulated. The orbit parameters were taken from Metop-A (orbits 4655, 4656, 6985 and 9744). Geophysical data for the time of the four orbits were taken from ERA5 (mate.copernicus.eu/climate-reanalysis). ERA5 lacks data on precipitation of convective nature. To compensate for this, precipitation was added from a separate database (provided by Alan Geer at ECMWF), based on similarities of large-scale precipitation profiles and some other variables. Using these data, radiances were simulated for all MWI and ICI channels, covering an area broader than the instrument's swath and representing a set of incidence angles.

These simulations were done by running the ARTS software in its three-dimensional mode (Eriksson et al., 2011). Absorption due to gases and liquid water content was calculated following Rosenkranz (1993, 1998) and Ellison (2007), and surface emissivities following Prigent et al. (2017) and Aires et al. (2011). The size distribution of rain drops and ice hydrometeors were set following Abel and Boutle (2012) and Field et al. (2007), respectively. Particle properties were taken from Eriksson et al. (2018), applying ID25 for rain and IDs 15 and 20 for ice hydrometeors (the name of these habits are found Table 2 below).

Using the set of pre-calculated pencil-beam radiances as a "look-up"-table, antenna weighted brightness temperatures could be generated with a relatively low calculation burden taking full account of MWI's and ICI's scanning and footprints

characteristics, for different assumptions of the exact syncing between the instruments. In all parts, the WGS-84 reference ellipsoid was applied.

**4.1.2 Main findings**

The assumption here is that the goal of the remapping is to obtain data as would be observed with a synthetic instrument having
5  a common footprint for all channels (implying the same surface incidence angle for all channels). As will be shown, this can not be achieved perfectly. However, these "errors" can at least partly be considered in the retrieval process and the final impact can be relatively low. Most importantly, the basic impact of different incidence angles can be included in both 1DVAR and BMCI retrievals.

[revised manuscript text omitted]

Observations over both water and land were simulated. Ocean surface emissivity was modelled according to Prigent et al. (2017). In lack of any model for ICI's frequency range, land emissivity was simply set to vary randomly around 0.9 (with a log-normal distribution). The data used below contain in total $6.2 \cdot 10^6$ cases, based on 1373 CloudSat orbits between Sep 2015 and Jan 2016.

5    Besides the retrieval database, a smaller dataset was also simulated for channels 16-21 of ATMS (Weng et al., 2012) and a statistical comparison to actual observations was made. The simulated data were generated exactly as done for the retrieval database, except that the footprint averaging followed the specifications of ATMS. Example results are displayed in Fig 7. The peak in the distribution around 255 K corresponds to "clear-sky" situations (low level cloud can still be present), while most cases below ~230 K should contain influences of ice hydrometeors. The agreement between simulations

10    and observations is high down to about 200 K. For lower brightness temperatures the simulations show higher occurrence rates than the observations. This deviation is at least partly a consequence of that the full antenna pattern and particle orientation are not yet considered in the simulations. The better agreement for nadir simulations, where ATMS has a smaller footprint, indicates the impact of the first of these two effects. By assuming totally random particle orientation radar back-scattering is under-estimated and our procedure will generate clouds with a high bias in IWC. There is a compensating effect when

15    simulating the passive data, by a similar under-estimation of extinction, but it is smaller, at least for angles away from nadir where particle orientation has a smaller impact on the projected cross-section (Brath et al., 2019).

The approach behind the database generation reproduces GMI (Draper et al., 2015) data in a similar manner, even when focusing on the tropical Pacific where deep convective systems control the impact of ice hydrometeors on ICI and the radiative transfer simulations are especially challenging (Ekelund et al., 2019).

20    A similar comparison is found in Fig. 13 of Geer and Baordo (2014). They obtained a poorer agreement with observations, with an underestimation starting at about 225 K. Similar particle models were used and the better agreement found here is likely a consequence of that the simulations are based on CloudSat, and not model data. The agreement is similar for the

other ATMS channels considered, see Rydberg (2018). A graphical manner for exploring if the retrieval database covers the multi-dimensional space spanned by the observations to be inverted is found in Brath et al. (2018, Fig. 2).

**4.3.2 Degrees of freedom**

As an introduction to the information provided by ICI, Fig. 8 displays an estimate of the measurements' degrees of freedom
5 ( DoF) for tropical conditions. The DoF can be seen as a measure on the effective number of channels.

Each DoF-value is calculated by finding the (left) eigenvectors ($\boldsymbol{E}$) of the simulated set of measurement vectors in consideration (without noise added). These eigenvectors and the covariance matrix ($\boldsymbol{S_y}$) of the data are related as:

$$\boldsymbol{S_y} = \boldsymbol{E}\boldsymbol{\Lambda}\boldsymbol{E}^T \tag{14}$$

where $\Lambda$ is a diagonal matrix, holding the eigenvalues. See e.g. Eriksson et al. (2002) for further details. The uncertainty due
10 to thermal noise, in the eigenvalue space, is

$$\boldsymbol{S_\Lambda} = \boldsymbol{E}\boldsymbol{S_\epsilon}\boldsymbol{E}^T, \tag{15}$$

where $\boldsymbol{S_\epsilon}$ has $\mathrm{NE\Delta T}^2$ as its diagonal elements and is zero elsewhere (cf. Eq. 13). As $\boldsymbol{S_\epsilon}$ is diagonal, also $\boldsymbol{S_\Lambda}$ will be diagonal due to properties of the eigenvectors (orthonormality). The number of diagonal elements in $\boldsymbol{S_y}$ that are larger than the corresponding value in $\boldsymbol{S_\Lambda}$ can be taken as the DoF. This calculation of DoF is essentially the same as the analysis described
15 in Sec. 2.4.1 of Rodgers (2000), but is somewhat more general as it is based on $\boldsymbol{S_y}$ and does not involve the Jacobian matrix, so it can be easily computed even in cases where Jacobians are not available.

For very low IWP and most wet atmospheres, the  DoF is only two. For these conditions, ICI is primarily sensitive to humidity in the middle and upper troposphere. The  DoF increases with decreasing IWV, as humidity at lower altitudes then gets a growing impact. The  DoF is here about three, consistent with the fact that ICI has three channels around each
20 water vapour transition covered (1V-3V, 5V-7V and 9V-11V, respectively), and that there is a high redundancy in information between these groups of channels (which together give an improved precision for water vapour retrievals). Figure 4 shows that the two innermost 448 GHz channels cover higher altitudes than the other channels, but it appears that these two channels add little information in single-footprint retrievals due to a relatively high noise (Table 1). A further analysis of ICI's overall performance for clear-sky conditions is left for a future study. For most dry atmospheres, there is also a surface contribution
25 to the  DoF, mainly by channels 4V and 4H, from the various variables affecting surface emission and reflectivity.

The  DoF is considerably higher at high IWP. The maximum  DoF in Fig. 8 is eight, but the true number is likely higher. The figure is based on simulations only including totally random particle orientation and thus the full information given by the dual polarisation channels is not reflected. The simulations lack also melting particles and still use a relatively low
30 number of particle models, and the full variability of hydrometeors is probably not yet reflected.

There is an intermediate range, extending between about 10 and $500\,\mathrm{g/m}^2$, where  DoF is increasing with IWP. This analysis shows that ICI acts mainly as a coarse humidity sounder for IWP below $\sim 10\,\mathrm{g/m}^2$, but, as designed, provides more

[revised manuscript text omitted]

**5 Outlook**

The basic algorithm will not be modified until some time after the launch of ICI and the main concern for the coming years is to refine the retrieval database generation. A required extension is to include particle orientation, as shown by Defer et al. (2014) and Gong and Wu (2017). The first data on scattering properties at sub-millimetre wavelengths of oriented particles have just been presented (Brath et al., 2019). Varying orientation distributions should be used in the database generation. Scattering solvers handling oriented particles include RT4 (Evans and Stephens, 1995b) and DOIT (Emde et al., 2004).

In the database used in this work, a strict separation between liquid and ice hydrometers was assumed. This is a simplification in several ways. Super-cooled liquid cloud droplets are common in the atmosphere (e.g. Zhang et al., 2010), frequently as part of "mixed-phase" clouds. Results in Pfreundschuh et al. (2019) indicate that ICI has some sensitivity to such super-cooled liquid water and it should thus be considered in future work. Also the super-cooled liquid water in updraft regions of convective cells should be taken into account, especially as the drops here can be of mm-size and the liquid water content can reach several g/m$^3$ (Lawson et al., 2015). This should lead to a significant impact on both CloudSat and ICI data. Finally, the impact of melting  ice hydrometeors should be assessed and  included if found relevant. However, data on single scattering properties of such particles are still lacking for the frequency range of ICI.

A broader range of particle size distributions and particle shapes should be used, compared to the simulations used in this work. 
[revised manuscript text omitted]

As a last remark we would like to stress that ICI will provide the first "operational" observations of our atmosphere in the sub-millimetre region and its data will cover more than 20 years. This will give the weather forecasting and climate communities a new important data source.

[revised manuscript text omitted]

**Figure 9.** Estimated retrieval performance for IWP (top panel), $Z_m$ (Eq. 4, middle panel) and $D_m$ (Eq. 5, bottom panel). Tropical refers to data at latitudes between $30°$S and $30°$N, while mid-latitude includes data for November to January  between $35° - 65°$N. The blue and yellow solid lines show the median of retrieved median value, while the corresponding dashed lines show the median of retrieved 5th and 95th percentile. The performance for $Z_m$ and $D_m$ is shown for states with an IWP above 25 and $50\,\text{g/m}^2$ for tropical and mid-latitude, respectively.